# Mix-CPT: A Domain Adaptation Framework via Decoupling Knowledge Learning and Format Alignment

**Jinhao Jiang[1*], Junyi Li[2*], Wayne Xin Zhao[1✉] , Yang Song[3✉], Tao Zhang[3] and Ji-Rong Wen[1]**
[1]Gaoling School of Artificial Intelligence, Renmin University of China.
[2]Department of Computer Science, National University of Singapore.
[3]Nanbeige Lab, BOSS Zhipin.
`jiangjinhao@ruc.edu.cn, batmanfly@gmail.com`

## Abstract

Adapting large language models (LLMs) to specialized domains typically requires domain-specific corpora for continual pre-training to facilitate knowledge memorization and related instructions for fine-tuning to apply this knowledge. However, this method may lead to inefficient knowledge memorization due to a lack of awareness of knowledge utilization during the continual pre-training and demands LLMs to simultaneously learn knowledge utilization and format alignment with divergent training objectives during the fine-tuning. To enhance the domain adaptation of LLMs, we revise this process and propose a new domain adaptation framework including domain knowledge learning and general format alignment, called *Mix-CPT*. Specifically, we first conduct a knowledge mixture continual pre-training that concurrently focuses on knowledge memorization and utilization. To avoid catastrophic forgetting, we further propose a logit swap self-distillation constraint. By leveraging the knowledge and capabilities acquired during continual pre-training, we then efficiently perform instruction tuning and alignment with a few general training samples to achieve format alignment. Extensive experiments show that our proposed *Mix-CPT* framework can simultaneously improve the task-solving capabilities of LLMs on the target and general domains.

## 1 Introduction

Large Language Models (LLMs) (Zhao et al., 2023) have revolutionized the field of natural language processing (NLP) (OpenAI, 2023), showing exceptional capabilities such as instruction following and complex reasoning (Wei et al., 2022). However, due to their limited exposure to relevant data, such general LLMs still considerably lag behind in specific domains requiring professional knowledge. This situation has necessitated the effective adaptation of general-purpose LLMs to specific domains (*e.g.,* mathematics and code), called *domain adaptation* of LLMs (Guo & Yu, 2022).

In essence, tailoring general LLMs to specific domains requires adaptation in two main aspects, namely *knowledge learning* (acquiring and leveraging the necessary domain knowledge) and *format alignment* (responding to the user in an expected output form) (Jiang et al., 2024; Zhou et al., 2023; Hu et al., 2024b). Specially, knowledge learning can be further fulfilled via knowledge memorization and utilization. In practice, domain adaptation of LLMs typically involves three consecutive stages (Rozière et al., 2023; Azerbayev et al., 2023), *i.e.,* pre-training, instruction tuning, and alignment, where the first stage is primarily aimed at knowledge memorization and the other two stages are mainly focused on knowledge utilization and format alignment. However, at the pre-training stage, knowledge memorization based on raw domain-specific corpora would be somehow inefficient without eliciting the acquired knowledge according to task goals (Jiang et al., 2024). Despite that some studies incorporate instruction data for pre-training, they often rely on proprietary models to synthesize high-quality instructions at scale (Cheng et al., 2024a; Wang et al., 2024), which may

---

[*] Equal contribution.
[✉] Corresponding author.

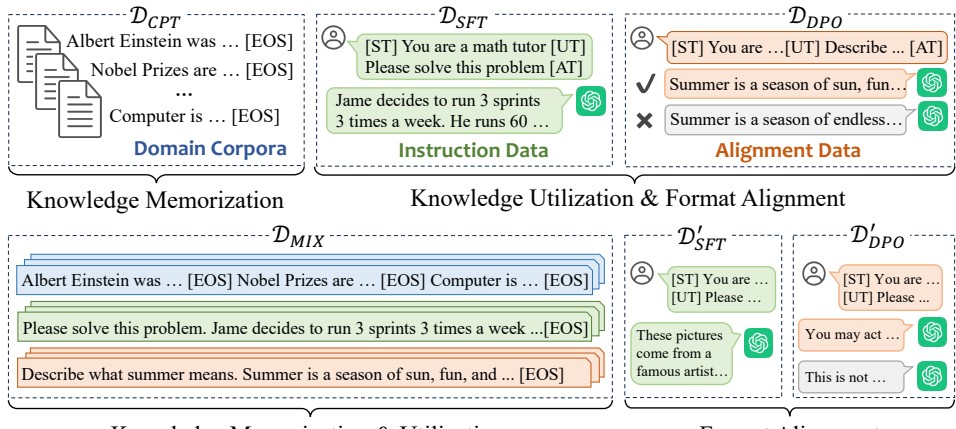

Figure 1: Comparison of traditional domain adaptation approaches (top) and our proposed Mix-CPT paradigm (bottom). "[EOS]" is a special token representing the end of a document. "[ST]", "[UT]", and "[AT]" denote the system, user, and assistant chat template, respectively.

not be that easy without extensive fine-tuning experiences. Another issue is that learning to master knowledge utilization and format alignment in both instruction tuning and alignment stages might lead to suboptimal performance, since the two goals can be divergent in model optimization (Ren et al., 2024).

Considering the above issues, this paper explores a new domain adaptation approach that only uses raw domain-specific corpora and general instruction or alignment data. Our hypothesis is that the knowledge utilization capacity can be essentially learned from general instruction or alignment data, which has also been evidenced by prior studies (Ouyang et al., 2022b). In this way, we can remove the tedious instruction synthesis step from the training pipeline, since it is much easier to obtain general or mixed domain instruction data from open resources. Another important attempt is to enhance knowledge learning by jointly acquiring both memorization and utilization of knowledge. Therefore, we schedule all the instruction and alignment data at the pre-training stage (with a suitable format), then only reuse a minor proportion of instruction and alignment data for fine-tuning to achieve format alignment. We give a comparison in Figure 1.

Specially, our approach for domain adaptation of LLMs consists of two main stages, *i.e., domain knowledge learning* and *general format alignment*. In the first stage, we conduct knowledge mixture continual pre-training (*Mix-CPT*) to integrate both knowledge memorization and utilization. In the second stage, based on the knowledge and capabilities that are already acquired during pre-training, we perform instruction tuning and alignment in an efficient manner to achieve format alignment. For continual pre-training, we convert raw domain documents, general instructions, and alignment data into a unified format. To avoid catastrophic forgetting, we propose Logit Swap Self-Distillation (*LSSD*), which exchanges the predicted top-1 token logit with the logit of the ground-truth token, serving as the surrogate target. In this way, LSSD maintains most probabilities of the original distribution of LLMs, thereby preserving original capabilities. In instruction tuning and alignment, we propose a novel format alignment score as criterion to select a small number of instructions from the pre-training instruction set. These instructions have already been seen during pre-training, so that the model can mainly focus on pure style or format learning for downstream tasks.

We conduct experiments on domain-specific and general tasks to verify the effectiveness of our Mix-CPT method, including a total of seven distinctive capabilities based on 17 representative benchmarks. For both base LLMs and chat LLMs, our approach can effectively improve their domain-specific and general performance compared to traditional methods of first performing continual pre-training, followed by instruction tuning and alignment.

## 2 APPROACH

To adapt general LLMs to specific domains (*e.g.,* wiki, code), our core idea is to decouple knowledge learning and format alignment, and propose an effective two-stage domain adaptation framework,

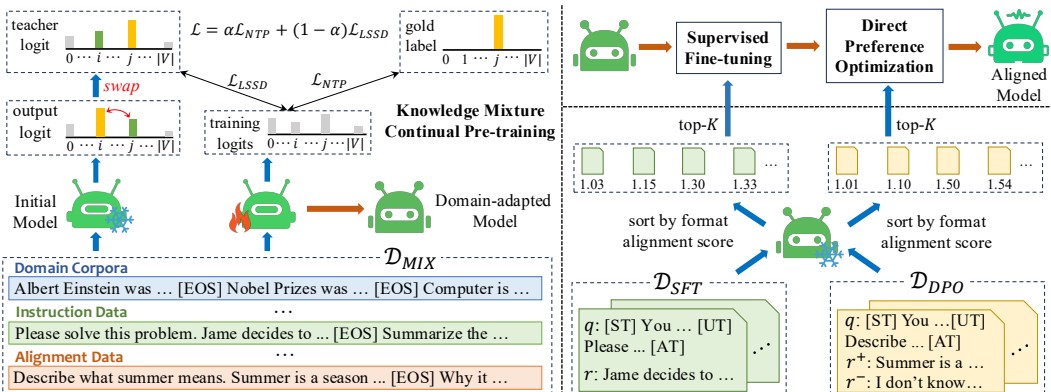

Figure 2: Overall framework of our proposed rescheduled domain adaptation paradigm, Mix-CPT. "[ST]", "[UT]", and "[AT]" denote the system, user, and assistant prompts, respectively.

*i.e.,* first performing knowledge mixture continual pre-training (Section 2.1) and then performing efficient format alignment (Section 2.2). We show the overall architecture in Figure 2.

## 2.1 KNOWLEDGE MIXTURE CONTINUAL PRE-TRAINING

Different from previous work that performs continual pre-training solely based on domain-specific corpora (Que et al., 2024; Ke et al., 2022), we propose to mix domain-specific documents, general instructions and alignment data as pre-training data. The QA-based instruction and alignment data is useful to reflect how the knowledge will be accessed and utilized through questions, enhancing the learning of new knowledge in domain-related documents. Besides, using general instructions can facilitate LLMs to transfer the general knowledge utilization capability to specific domains without relying on domain-related instructions in prior work (Jiang et al., 2024; Cheng et al., 2024a). Specifically, we first transform raw domain documents, general instructions, and alignment data into a unified format for continual pre-training. To avoid catastrophic forgetting, we further introduce a logit swap self-distillation (LSSD) approach during the continual pre-training process.

### 2.1.1 UNIFIED KNOWLEDGE FORMAT

Typically, adapting LLMs to a specific domain involves three distinct and relatively independent stages, each based on corresponding data in different formats. Specifically, the base model firstly performs continual pre-training (CPT) on domain-specific corpora for learning new knowledge, then conducts supervised fine-tuning (SFT) based on instructions for enhancing the instruction following ability, and finally utilizes the preference data for human alignment. In this work, we adopt direct preference optimization (DPO) as the alignment algorithm. Formally, we denote the domain-specific corpus as $\mathcal{D}_{\text{CPT}} = \{d_i\}_{i=1}^{n_c}$, where $d_i$ represents a raw domain document consisting of a sequence of tokens. For the instructions used in SFT, we denote as $\mathcal{D}_{\text{SFT}} = \{\langle q_i, r_i \rangle\}_{i=1}^{n_s}$, where $q_i$ and $r_i$ represent the user query and the expected response, repectively. For alignment data used in DPO, we denote by $\mathcal{D}_{\text{DPO}} = \{\langle q_i, r_i^+, r_i^- \rangle\}_{i=1}^{n_d}$, where $q_i$, $r_i^+$, and $r_i^-$ represent the user query, positive response, and negative response, respectively.

In this paper, we propose to mix $\mathcal{D}_{\text{CPT}}$, $\mathcal{D}_{\text{SFT}}$ and $\mathcal{D}_{\text{DPO}}$ into a mixture dataset $\mathcal{D}_{\text{MIX}}$ with a unified text format, building upon which we further perform knowledge mixture continual pre-training on a general LLM. Unlike previous work relying on synthesizing high-quality domain instructions (Jiang et al., 2024; Cheng et al., 2024a), we empirically find that knowledge utilization is actually a general capability that can be learned from general instructions, and we can further transfer such ability to enhance the learning of domain knowledge. Specially, we remove any templates (*e.g.,* [UT]) from instructions and alignment data to construct the mixture data, which consists of three kinds of samples denoted by $\mathcal{D}_{\text{MIX}} = \{x_{\text{cpt}}, x_{\text{sft}}, x_{\text{dpo}}\}$, where $x_{\text{cpt}} = d_i$ is the original domain document, $x_{\text{sft}} = [q_i; r_i]$ denotes the concatenation of user query and expected response in the instruction, and $x_{\text{dpo}} = [q_i; r_i^+]$ is the concatenation of user query and positive response in the alignment data.

Following existing pre-training methods (Touvron et al., 2023), we concatenate samples within each kind of data (*i.e.,* $\mathcal{D}_{\text{CPT}}$, $\mathcal{D}_{\text{SFT}}$ and $\mathcal{D}_{\text{DPO}}$) and truncate the sequence when reaching the maximum

input length of the LLM. Besides, we add an extra special symbol (*i.e.,* `[EOS]`) at the end of each sample to separate them. We repeat this process until concatenating all samples to obtain our final knowledge mixture pre-training data $\mathcal{D}_{\mathrm{MIX}}$.

Note that though we use general instruction data to derive the mixture data here, our approach can be generally extended to incorporating domain-specific instruction data (Cheng et al., 2024a), which often relies on specific data synthesis techniques.

### 2.1.2 LOGIT SWAP SELF-DISTILLATION

After obtaining the unified mixture data $\mathcal{D}_{\mathrm{MIX}} = \{x_{\mathrm{cpt}}, x_{\mathrm{sft}}, x_{\mathrm{dpo}}\}$, we then perform continual pre-training on the base LLM. For simplicity, we remove the subscript of each training sample in the mixture data, denoted as $x$. We adopt the pre-training task of next token prediction (NTP). Specifically, given an input $x = \{w_1, w_2, ..., w_n\}$, we adopt the standard language modeling objective to minimize the cross-entropy loss as follows:

$$\mathcal{L}_{\mathrm{NTP}} = -\sum_{j=1}^{n} \log \Pr(w_j | w_{<j}; \Theta), \tag{1}$$

where $w_j$ denotes the $j$-th token in the input, $w_{<j}$ is the previous tokens, and $\Theta$ denotes the model parameters. During continual pre-training, the task of next-token prediction enables the base LLM to learn domain knowledge, and the incorporation of general instructions and the alignment data further transfers the general knowledge utilization capability to specific domains.

However, the traditional language modeling objective is prone to suffer from the issue of catastrophic forgetting for previously learned knowledge of LLMs. Therefore, we propose an auxiliary training objective, *i.e., Logit Swap Self-Distillation (LSSD)*, which serves as an extra constraint for the standard language modeling objective. Specifically, we first utilize the original base LLM before continual pre-training (paramerized by $\Theta_{\mathrm{ori}}$) to infer the output logits following the standard language modeling objective yet without computing the loss:

$$\boldsymbol{h}_j = \mathrm{LLM}(w_{<j}; \Theta_{\mathrm{ori}}), \tag{2}$$

$$\boldsymbol{l}_j = \boldsymbol{h}_j \boldsymbol{W}_e^T, \tag{3}$$

where $\boldsymbol{W}_e$ is the token embedding matrix, $\boldsymbol{h}_j$ is the hidden state of the last transformer block, and $\boldsymbol{l}_j$ denotes the output logit at the $j$-th position. Then, we exchange the logit value of the top-1 predicted token (*i.e.,* $\widetilde{w}_j$) and the ground-truth token (*i.e.,* $w_j$) if they are not equal:

$$\tilde{\boldsymbol{l}}_j = \mathrm{Exchange}(\boldsymbol{l}_j, I_{\widetilde{w}_j}, I_{w_j}) \quad \text{if} \ \ I_{\widetilde{w}_j} \neq I_{w_j}, \tag{4}$$

where $I_{\widetilde{w}_j}$ and $I_{w_j}$ are indices of $\widetilde{w}_j$ and $w_j$ in the vocabulary respectively, the function $\mathrm{Exchange}(\cdot)$ will exchange their logit values in $\boldsymbol{l}_j$. The swapped logit $\tilde{\boldsymbol{l}}_j$ will be regarded as the teacher logit in LSSD. In essence, LSSD only calibrates the prediction of ground-truth token for adapting to the current domain knowledge while maintaining most originally learned knowledge of the LLM (*i.e.,* represented by the unchanged logit values in $\tilde{\boldsymbol{l}}_j$). Next, we can compute the teacher model's probability distribution for the $j$-th token with softmax function:

$$\Pr(w_j | w_{<j}; \Theta_{\mathrm{ori}}) = \mathrm{softmax}(\tilde{\boldsymbol{l}}_j). \tag{5}$$

Finally, we compute the self-knowledge distillation objective and minimize the reverse Kullback-Leibler divergence loss (Gu et al., 2023) between the current model's probability distribution and the teacher model's probability distribution as follows:

$$\mathcal{L}_{\mathrm{LSSD}} = -\sum_{j=1}^{n} \Pr(w_j | w_{<j}; \Theta) \log(\frac{\Pr(w_j | w_{<j}; \Theta)}{\Pr(w_j | w_{<j}; \Theta_{\mathrm{ori}})}). \tag{6}$$

In the knowledge mixture continual pre-training stage, the final total loss is the combination of next token prediction loss and self-distillation loss, controlled by a co-efficient $\alpha$ as follows:

$$\mathcal{L}_{\mathrm{CPT}} = \alpha \cdot \mathcal{L}_{\mathrm{NTP}} + (1 - \alpha) \cdot \mathcal{L}_{\mathrm{LSSD}}. \tag{7}$$

Due to the significant distributional differences between domain corpora and original training data, traditional approach of using one-hot label can lead to substantial model updates, which may cause catastrophic forgetting of the original knowledge after adaptation. In contrast, LSSD adopts the prediction from the original model as a surrogate label by only modifying the logit of top-1 token. This way can effectively maintain most previously learned knowledge but adapt to new domains.

## 2.2 GENERAL FORMAT ALIGNMENT

In the domain knowledge learning stage, the LLM has simultaneously learned to memorize domain knowledge and understand how to utilize the knowledge through our proposed knowledge mixture continual pre-training. After that, during the format alignment stage, the LLM can be more *efficiently* fine-tuned to master the task format with only a small number of alignment samples. Next, we first introduce the selection of training samples and then perform efficient format alignment.

**Alignment Sample Selection.** Since we would like to decouple knowledge learning and format alignment, we aim to select training samples from $\mathcal{D}_{\text{SFT}}$ and $\mathcal{D}_{\text{DPO}}$ that have been *encountered* during continual pre-training, which can avoid introducing new knowledge in subsequent fine-tuning. The selection criterion is based on the *format alignment score (FAS)*, which estimates the difficulty of a given SFT sample $\langle q_i, r_i \rangle$ by comparing the conditional loss with and without formatted instruction:

$$\text{FAS}(\langle q_i, r_i \rangle) = \frac{\mathcal{L}_{\text{NTP}}(r_i | \hat{q}_i)}{\mathcal{L}_{\text{NTP}}(r_i)}, \tag{8}$$

where $\hat{q}_i$ denote the formatted query equipped with chat templates for interaction with humans. For a DPO sample $\langle q_i, r_i^+, r_i^- \rangle$, FAS compares the conditional losses of positive response and negative response given the formatted query:

$$\text{FAS}(\langle q_i, r_i^+, r_i^- \rangle) = \frac{\mathcal{L}_{\text{NTP}}(r_i^+ | \hat{q}_i)}{\mathcal{L}_{\text{NTP}}(r_i^- | \hat{q}_i)}. \tag{9}$$

In essence, high FAS scores infer the difficulty of generating responses given a query with an interaction format and the significant disparity between positive and negative ones. Therefore, we finally select top-$K$ samples with the highest FAS scores to conduct supervised fine-tuning and direct preference optimization.

**Efficient Format Alignment.** After selecting very few training samples, we can utilize them to conduct format alignment in an efficient manner. Firstly, we utilize the selected instruction samples from $\mathcal{D}_{\text{SFT}}$ to perform supervised fine-tuning following the standard way (Ouyang et al., 2022b), which is to minimize the cross-entropy loss:

$$\mathcal{L}_{\text{SFT}} = -\sum_{j=1}^{n} \log \text{Pr}(r_j | q, r_{<j}; \Theta), \tag{10}$$

where $r_j$ and $r_{<j}$ denote the $j$-th token and its previous tokens in the response. Secondly, we utilize the selected preference samples from $\mathcal{D}_{\text{DPO}}$ to conduct direct preference optimization following Rafailov et al. (2023b) as follows:

$$\mathcal{L}_{\text{DPO}} = -\log \sigma \left( \beta \log \frac{\pi(r^+ | q; \Theta)}{\pi(r^+ | q; \Theta_{\text{ref}})} - \beta \log \frac{\pi(r^- | q; \Theta)}{\pi(r^- | q; \Theta_{\text{ref}})} \right), \tag{11}$$

where $\sigma$ denotes the sigmoid function, $\Theta_{\text{ref}}$ denotes the parameters of the reference LLM (usually the original LLM itself), and $\pi$ denotes the product of the probabilities of all output tokens.

## 3 EXPERIMENTS

### 3.1 EXPERIMENTAL SETUP

In our experiments, we mainly focus on three popular domains for adapting general LLMs, *i.e.,* *encyclopedia*, *mathematics*, and *code*. And select official Wikipedia dump[1], AutoMathText (Zhang et al., 2024), and StarCoder (Li et al., 2023) as the corresponding domain-specific corpus. For general instruction datasets, we choose *TULU-V2-mix* (Ivison et al., 2023) and *UltraFeedback* (Cui et al., 2023) for instruction tuning and alignment, respectively. We employ two prominent LLMs as the base model, *i.e., QWen2-7B* (Yang et al., 2024) and Meta-Llama-3-8B (Dubey et al., 2024). For comparative analysis, we consider *official chat LLMs*, *reimplemented chat LLMs*, and *continual pre-training augmented LLMs*. We evaluate seven distinctive capabilities of LLMs in a total of 17 representative NLP datasets based on the OpenCompass platform (Contributors, 2023). We show the details of our complete experimental setup in Appendix A.

---

[1]https://dumps.wikimedia.org/enwiki/20240301/

Table 1: Evaluation results on three specialized domains and two general examinations. The underline and **bold** fonts denote the best results in the target domain and the average results in each domain adaptation group, respectively.

| | Model | Wiki | Math | Code | English Examination | Chinese Examination | Average |
|---|---|---|---|---|---|---|---|
| | LLaMA3-8B-Base | 53.81 | 33.62 | 42.31 | 51.02 | 46.69 | 45.92 |
| *Wiki* | + CPT | 54.20 | 34.07 | 37.42 | 52.97 | 46.07 | 45.64 |
| | + Mix-CPT (*w/o KD*) | 55.81 | 38.96 | 41.62 | 52.19 | 47.44 | 47.68 |
| | + Mix-CPT | 55.47 | 37.91 | 42.37 | 52.55 | 49.93 | **47.91** |
| *Math* | + CPT | 54.14 | 34.80 | 28.95 | 50.47 | 49.02 | 43.62 |
| | + Mix-CPT (*w/o KD*) | 55.55 | 37.99 | 42.43 | 46.16 | 48.60 | 45.90 |
| | + Mix-CPT | 55.11 | 37.71 | 45.14 | 51.54 | 49.89 | **48.04** |
| *Code* | + CPT | 50.98 | 34.19 | 35.64 | 51.14 | 47.92 | 44.29 |
| | + Mix-CPT (*w/o KD*) | 54.93 | 39.38 | 42.35 | 46.31 | 47.91 | 46.02 |
| | + Mix-CPT | 55.50 | 38.15 | 42.82 | 51.55 | 48.50 | **47.61** |
| | QWen2-7B-Base | 46.37 | 62.33 | 60.02 | 46.56 | 82.90 | 56.00 |
| *Wiki* | + CPT | 47.20 | 59.48 | 29.45 | 52.32 | 81.92 | 51.12 |
| | + Mix-CPT (*w/o KD*) | 46.65 | 58.54 | 58.67 | 48.03 | 80.86 | 55.27 |
| | + Mix-CPT | 46.88 | 59.77 | 62.06 | 48.78 | 81.84 | **56.56** |
| *Math* | + CPT | 46.22 | 58.60 | 48.13 | 44.42 | 82.51 | 52.17 |
| | + Mix-CPT (*w/o KD*) | 46.19 | 59.42 | 59.93 | 46.12 | 82.70 | 55.21 |
| | + Mix-CPT | 46.42 | 60.58 | 61.40 | 45.60 | 82.25 | **55.58** |
| *Code* | + CPT | 41.37 | 60.02 | 41.93 | 42.05 | 82.97 | 49.58 |
| | + Mix-CPT (*w/o KD*) | 45.29 | 59.40 | 58.53 | 48.36 | 81.68 | 55.32 |
| | + Mix-CPT | 45.50 | 59.20 | 60.21 | 48.18 | 82.94 | **55.73** |

## 3.2 MAIN RESULTS

Table 1 and Table 2 display the evaluation results for base and chat LLMs respectively, comparing our proposed Mix-CPT method and other baselines.

### 3.2.1 RESULTS OF BASE LLMs

We initially assess the effectiveness of our proposed Mix-CPT framework for base LLMs in mitigating catastrophic forgetting and facilitating the knowledge learning during continual pre-training, especially the logit swap self-distillation constraint. To better observe the impact of knowledge mixture continual pre-training on the performance in target domains and general capabilities, we select domain-specific tasks and general examination tasks. We show the evaluation results in Table 1.

First, we can see that traditional continual pre-training (*i.e.,* + CPT) does not necessarily enhance the performance of base LLMs in the target domain, and may instead impair their performance therein. Besides, this method inevitably leads to a certain degree of catastrophic forgetting, thereby damaging the overall performance of the base LLM. For example, compared to the LLaMA3-8B-Base model, typical CPT leads to observed improvements in math (*i.e.,* $33.62 \rightarrow 34.80$) but diminishes overall performance (*i.e.,* $45.92 \rightarrow 43.62$). The phenomenon has been reported in existing work (Lin et al., 2024). We notice that the performance of the QWen2-7B-Base in math and code domains declines after undergoing either traditional continual pre-training or Mix-CPT. We speculate that this may be due to the significant discrepancy between the distribution of the continual pre-training data and the data used in the final stage of original pre-training (*e.g.,* annealing) (Yang et al., 2024; Dubey et al., 2024). Besides, the model might have already been trained on these two datasets, and further training leads to overfitting. For example, the typical CPT method shows a decline in the math domain (*i.e.,* $62.33 \rightarrow 58.60$) and the overall result (*i.e.,* $56.00 \rightarrow 52.17$). By contrast, Mix-CPT significantly alleviates such a decline and better maintains general performance (*i.e.,* $56.00 \rightarrow 55.58$). This demonstrates that our framework can effectively mitigate the impact of knowledge forgetting caused by traditional incremental training processes. This is highly beneficial in real-world scenarios where the training data proportion and distribution of the model are usually unknown.

Table 2: Evaluation results on three specialized domains and four general capabilities (*i.e.,* Reading Comprehension, Complex Reasoning, EXamination, and Instruction Following). The "CSD" denotes the typical domain adaptation method of conducting CPT, SFT, and DPO. The underline and **bold** fonts denote the same meaning as Table 1.

| Model | | | Specialized Domain | | | General Domain | | | | Average |
|---|---|---|---|---|---|---|---|---|---|---|
| | | | Wiki | Math | Code | RC | CR | EX | IF | |
| LLaMA3 8B-Base | Official Chat | | 46.97 | 50.88 | 63.29 | 77.03 | 78.21 | 62.02 | 80.40 | 64.96 |
| | Reimplemented Chat | | 33.72 | 30.88 | 41.96 | 74.19 | 77.42 | 57.62 | 69.55 | 55.86 |
| | *Wiki* | + CSD | 28.27 | 30.67 | 43.47 | 72.61 | 73.74 | 55.82 | 66.51 | 53.82 |
| | | + Mix-CPT (ours) | 33.21 | 30.21 | 47.22 | 72.60 | 71.36 | 58.65 | 68.88 | **55.25** |
| | *Math* | + CSD | 27.88 | 39.48 | 50.95 | 74.89 | 76.90 | 60.90 | 69.89 | 58.16 |
| | | + Mix-CPT (ours) | 32.12 | 39.40 | 51.21 | 73.36 | 77.00 | 60.68 | 71.81 | **58.61** |
| | *Code* | + CSD | 27.64 | 39.27 | 51.75 | 75.04 | 77.44 | 62.34 | 70.88 | **58.75** |
| | | + Mix-CPT (ours) | 31.98 | 38.30 | 52.67 | 73.33 | 74.86 | 60.71 | 72.11 | 58.25 |
| QWen2 7B-Base | Official Chat | | 37.39 | 67.05 | 72.31 | 85.79 | 82.08 | 73.48 | 82.63 | 71.74 |
| | Reimplemented Chat | | 28.10 | 58.12 | 61.22 | 84.35 | 81.86 | 71.08 | 75.40 | 66.80 |
| | *Wiki* | + CSD | 30.29 | 56.75 | 61.78 | 82.61 | 81.66 | 71.23 | 75.92 | 66.79 |
| | | + Mix-CPT (ours) | 34.55 | 56.74 | 64.66 | 83.93 | 80.61 | 72.66 | 74.73 | **67.94** |
| | *Math* | + CSD | 28.58 | 56.78 | 60.85 | 83.11 | 81.83 | 70.79 | 75.87 | 66.45 |
| | | + Mix-CPT (ours) | 33.81 | 57.48 | 67.38 | 83.29 | 80.88 | 72.44 | 76.27 | **68.29** |
| | *Code* | + CSD | 29.41 | 56.87 | 62.74 | 83.58 | 81.77 | 70.94 | 76.79 | 66.94 |
| | | + Mix-CPT (ours) | 36.05 | 55.07 | 64.84 | 83.42 | 80.57 | 73.65 | 76.77 | **68.24** |

Second, when mixing the domain raw data with the additional instructions and alignment data (*i.e.,* Mix-CPT w/o KD), the domain-specific capability can be further improved, which indicates that the mixed instruction data can benefit the learning of the domain knowledge during continual pre-training. At the same time, it can mitigate the effect of other general capabilities and reduce the degradation of the overall performance. For example, compared to the traditional CPT, mixing domain corpus with additional instruction data (Mix-CPT w/o KD) can almost consistently improve the domain capability and overall average performance (*i.e.,* $41.93 \rightarrow 58.53$ and $49.58 \rightarrow 55.32$ for the QWen2-7B-Base model in the target Code domain and overall average performance respectively).

Finally, through applying the logit swap self-distillation strategy to the knowledge mixture continual pre-training process (*i.e.,* Mix-CPT), we can further reduce the impact on the pre-learned knowledge while maintaining the domain capability improvement, thereby mitigating the degradation of the general capabilities of LLMs. Therefore, these results demonstrate that the Mix-CPT framework with the logit swap self-distillation constraint can indeed promote knowledge learning and alleviate the issue of catastrophic forgetting to some extent.

### 3.2.2 RESULTS OF CHAT LLM

Subsequently, we assess the performance of final chat LLMs after instruction tuning and alignment for achieving format alignment. We show the evaluation results in Table 2.

First, the typical domain adaptation method of conducting CPT, SFT, and DPO (called CSD) faces challenges in simultaneously enhancing domains-specific capabilities while preserving general capabilities, in contrast to reimplemented chat models that do not utilize domain-specific raw data. For example, compared to reproduced LLaMA3-8B chat model, traditional CSD method not only decreases the factual question answering performance in Wiki (*i.e.,* $33.72 \rightarrow 28.27$) but also hurts the average performance (*i.e.,* $55.86 \rightarrow 53.82$). This phenomenon indicates that conventional domain adaptation methods may cause catastrophic forgetting and merely focus on knowledge memorization without considering how to utilize them, which might suffer from a memorization trap.

Second, with the same instruction data and alignment data, our method can successfully improve the performance on the target domain while maintaining the general capability compared to the reimplemented models. Compared to the reimplemented chat LLM with large-scale closed-source pre-training data but using the same instruction and alignment data, our proposed method can achieve

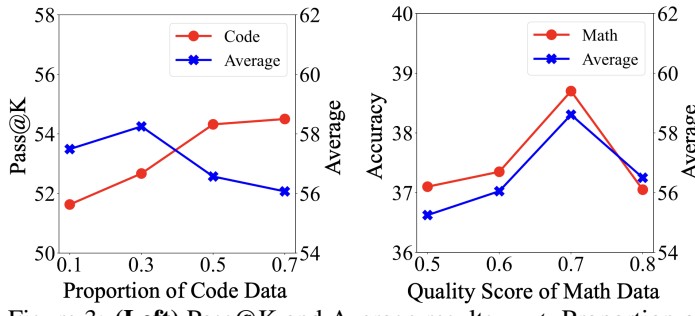 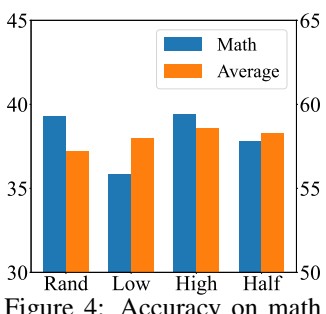

Figure 3: **(Left)** Pass@K and Average results *w.r.t.* Proportion of code data. **(Right)** Accuracy and Average results *w.r.t.* Quality score of math data.

Figure 4: Accuracy on math and Average results *w.r.t.* selection strategy.

better performance on the target domain (*e.g.,* 41.96 → 52.67 in the code domain for LLaMA3-8B-Base), which indicates the effectiveness of our method.

Finally, our proposed Mix-CPT method can simultaneously improve the performance of the target domains and the general capability. The main reasons are two fold. On one hand, based on the logit swap self-knowledge distillation constraint, the LLM can effectively memorize the raw domain data while maintaining its originally learned knowledge. On the other hand, by mixing the raw domain data with the general instructions and alignment data (removing any templates), the model can learn the general knowledge utilization capability and transfer to specific domains. In this way, the model can perform efficient format alignment with only a few formatted samples to better utilize both target domain knowledge and other general knowledge.

## 3.3 DETAILED ANALYSIS

In this section, we conduct a detailed analysis of the proposed method with LLaMA3-8B, using the same benchmarks as those presented in Table 2.

**Effect of Quantity and Quality of Raw Domain Data.** We first utilize the StarCoder dataset to explore the effect of the number of domain documents on domain-specific and general capabilities with consistent quality. Then, we adopt the AutoMathText dataset to explore the effect of the quality of raw domain data on LLMs' performance by leveraging the annotated quality scores. Specifically, we conduct two group experiments as follows:

• *Proportion of Code Data*: This group aims to compare the variants using different proportions of the StarCoder, including 10%, 30%, 50%, and 70%, while maintaining constancy in other variables.

• *Quality of Math Data*: This group aims to compare variants by employing various AutoMath-Text, each characterized by distinct quality scores with thresholds exceeding 0.5, 0.6, 0.7, and 0.8 respectively, while maintaining constancy in other variables.

We show the results in Figure 3. We can see that increasing the amount of raw domain documents can indeed further enhance the target domain performance under the same quality. However, even though using a larger number of raw domain data (*e.g.,* more math texts with $\geq 0.5$ score than those with $\geq 0.7$ score), the low quality of raw data can also decrease the LLMs' performance regardless of the target domain or general domain, which indicates the quality of domain data is a priority over its amount when performing continual pre-training. The final decline is due to that there are much less math texts with $\geq 0.8$ score (*i.e.,* only $10\%$ of math texts with $\geq 0.7$ score).

**Effect of Format Alignment Data Selection.** We conduct a further ablation study to explore the impact of different sample selection strategies for format alignment data, which consists of the difficulty and amount of selected data. Specifically, we conduct two groups of experiments including:

• *Difficulty of Samples for SFT and DPO*: This group compares four distinct selection strategies: random selection (Rand), easiest samples with the lowest FAS scores (Low), hardest samples with the highest FAS scores (High), and half easiest samples and half hardest samples (Half).

Table 3: Evaluation results on medicine and general domains with LLaMA3-8B-Base model when adapting to medicine domain. The underline and **bold** fonts denote the same meaning as Table 1.

| Model | Medicine Domain | | | | General Domain | | | | |
|---|---|---|---|---|---|---|---|---|---|
| | RCT | PubMedQA | MQP | Avg | MATH | MBPP | MMLU | CEVAL | Avg |
| LLaMA3-8B-Base | 73.6 | 59.8 | 66.2 | 60.1 | 18.0 | 57.9 | 65.9 | 49.3 | **47.8** |
| + CPT | 70.6 | 56.7 | 55.4 | 55.7 | 5.9 | 57.0 | 63.2 | 48.6 | 43.7 |
| + Mix-CPT | 71.0 | 63.4 | 79.2 | 63.4 | 16.6 | 58.0 | 66.2 | 49.7 | 47.6 |
| + Mix-CPT (Half) | 71.2 | 64.1 | 83.0 | **64.7** | 16.0 | 57.7 | 65.5 | 50.3 | 47.4 |

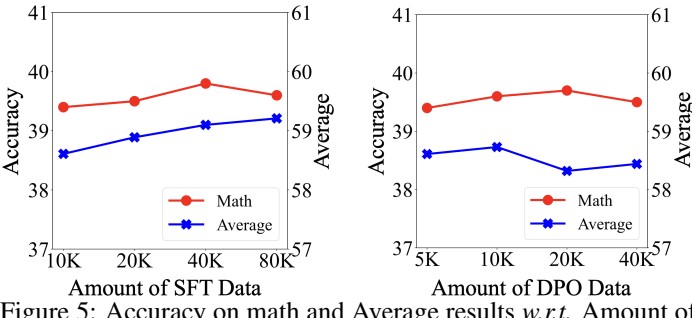

Figure 5: Accuracy on math and Average results *w.r.t.* Amount of SFT data (**Left**) and Amount of DPO data (**Right**).

Figure 6: Accuracy on Math and Average results *w.r.t.* distillation and normalization.

• *Amout of Samples for SFT and DPO*: This involves four variants using different quantities of easiest samples from the TULU-V2-mix dataset (*i.e.,* 10K, 20K, 40K, and 80K) and from the Ultra-Feedback dataset (*i.e.,* 5K, 10K, 20K, and 40K).

We show the results in Figure 4 and Figure 5. Firstly, we can see that, by using the samples with the highest FAS scores can balance the domain and general capability best compared to other selection strategies. Then, it enhances both domain and general abilities simultaneously to a certain extent by increasing the amount of SFT training samples. Conversely, when increasing the amount of DPO training samples, the results remain fluctuating, which might be due to the alignment tax.

**Effect of Knowledge Distillation.** To examine the effectiveness of our proposed logit swap self-distillation (LSSD), we compare our method to two distillation and normalization methods:

• *w/ LSSD:* this is our Mix-CPT model using LSSD.

• *w/ KD:* this replaces LSSD with a typical knowledge distillation method without logits swapping.

• *w/ L2:* this replaces LSSD with L2 normalization.

The results are shown in Figure 6. We can see that compared to directly using *L2* regularization to constrain the changes in model parameter weights, *KD* method can better preserve general capabilities when learning domain capabilities. Through logits swapping, our proposed *LSSD* method refines the inconsistencies with the gold label in *KD* method, further enhancing the learning of domain-specific capabilities. Finally, *LSSD* achieves the best balance among the three methods.

**Performance on Medicine Domain.** Here, we further explore the performance of Mix-CPT and traditional CPT in the medical domain. Specifically, following the existing work (Cheng et al., 2024b), we select the PubMed dataset as the domain-specific corpus and choose RCT (Dernoncourt & Lee, 2017), PubMedQA (Jin et al., 2019), and MQP McCreery et al. (2020) as the domain-related evaluation collections. For the general domain evaluation benchmarks, we select MATH, MBPP, MMLU, and CEVAL from the aforementioned experimental settings. We show the results in Table 3. We observe that compared to traditional CPT methods, our proposed Mix-CPT approach more effectively facilitates the learning of domain-specific skills and better maintains the general capabilities, indicating the effectiveness and generalizability of our method.

**Computational Efficiency of Mix-CPT.** We have discussed the additional overhead of our method in Appendix A.5. Here, we further conduct additional experiments by only randomly employing 50% pre-training data for performing the logit swap self-distillation, *i.e.,* Mix-CPT (Half), which

further reduce half of the costs introduced by self-distillation. We conduct the experiment on the medical domain with the LLaMA3-8B-Base model and evaluate its performance on domain-specific and general tasks as before. We show the results in Table 3. We find that compared to distilling the entire dataset, reducing the data by half further enhances the model's performance on domain-specific tasks, with only a slight decrease in general capabilities. This suggests that exploring even lower distillation ratios in the future could achieve a greater tradeoff between cost and performance.

## 4 RELATED WORK

**Domain Adaptation of LLMs.** Our work is closely related to efforts in adapting general LLMs to specific domains (Yildiz et al., 2024; Ke et al., 2022; Scialom et al., 2022). Due to the increasing scale and complexity of LLMs, training domain-specific LLMs from scratch involves significantly high financial and ecological costs (Luccioni et al., 2023). To address this issue, recent work has been devoted to studying efficient approaches like continual pre-training, which involves incrementally training general LLMs based on new domain corpora (Que et al., 2024; Ke et al., 2022), and continual fine-tuning, aiming to fine-tune general LLMs on a series of downstream tasks related to target domains (Razdaibiedina et al., 2023; Scialom et al., 2022; Luo et al., 2023a). However, these approaches might result in catastrophic forgetting and performance degradation in general language tasks (Kar et al., 2022; Mehta et al., 2023). Another line of work has explored conducting instruction fine-tuning by synthesizing domain-related instructions (Cheng et al., 2024a; Jiang et al., 2024). Nevertheless, these studies require additional models to synthesize amounts of instructions highly related to specific domains, resulting in high computational costs. It is noted that our method differs from these works in several ways. Firstly, we disentangle domain adaptation into knowledge memorization and capability elicitation, focusing on learning domain-specific knowledge and solving domain tasks with learned knowledge, respectively. Secondly, we employ token swap self-distillation in the mixture pre-training to retain general knowledge and avoid catastrophic forgetting.

**Instruction Tuning and Alignment.** Instruction tuning (also known as supervised fine-tuning) employs human-annotated instructions (Sanh et al., 2022; Mishra et al., 2022; Köpf et al., 2023; Sun et al., 2023) or synthetic instructions by proprietary models (Taori et al., 2023; Chiang et al., 2023; Wang et al., 2023) to fine-tune LLMs. Besides, alignment with reinforcement learning from human feedback (RLHF) (Ouyang et al., 2022a) or direct preference optimization (DPO) (Rafailov et al., 2023a) aims to align LLMs with human preference. Both instruction tuning and alignment are able to elicit knowledge from LLMs and improve their capabilities to solve downstream tasks. Recent work (Zhou et al., 2023) has demonstrated that LLMs mainly learn the style or format for interacting with users through simple instruction tuning and alignment, by leveraging their prior knowledge and capabilities already acquired during the pre-training stage. Furthermore, by comparing the token distribution before and after alignment, recent work (Lin et al., 2023) found that the most significant distribution shifts appear dominantly in stylistic tokens such as transitional phrases and discourse markers instead of contextual words that involve rich knowledge for solving downstream tasks. Inspired by these studies, we propose to expose knowledge memorization and capability elicitation from instruction tuning and alignment. Unlike these studies which typically focused on instruction tuning or alignment, we differ in that we unify the three stages of training LLMs (*i.e.,* continual pre-training, instruction tuning, and alignment) and conduct a knowledge mixture pre-training to mainly focus on learning new domain knowledge while maintaining general knowledge.

## 5 CONCLUSION

In this work, we proposed a two-stage domain adaptation approach, termed Mix-CPT, which encompasses both domain knowledge learning and general format alignment. Mix-CPT employed knowledge mixture continual pre-training to learn domain knowledge by integrating domain-specific raw data with general instructions and alignment data. Besides, we proposed Logit Swap Self-Distillation (LSSD) to relieve catastrophic forgetting. Based on the knowledge and capabilities acquired during pre-training, we then selected a small number of easy instructions to make the LLM to learn the style or format for interacting with human. Extensive experiments on three benchmark datasets showed that our proposed Mix-CPT outperforms the traditional method, obtaining improvements on both the domain and general capabilities.

## ACKNOWLEDGMENTS

This work was partially supported by National Natural Science Foundation of China under Grant No. 92470205, Beijing Municipal Science and Technology Project under Grant No. Z231100010323009, and Beijing Natural Science Foundation under Grant No. L233008. Xin Zhao is the corresponding author.

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

## APPENDIX

## A EXPERIMENTAL SETUP

### A.1 DOMAIN-SPECIFIC CORPUS

In our experiments, we mainly focus on three popular domains for adapting general-purpose LLMs, *i.e.,* encyclopedia, mathematics, and code. For the encyclopedia domain, we select Wikipedia as the primary corpus, which is collaboratively developed by volunteers globally and can be freely accessed online. To enable LLMs to learn knowledge from new documents, we utilize the official 2024/03/01 Wikipedia dump[2] and conduct necessary data cleaning and filtering processes such as deduplication, resulting in approximately 4B tokens in raw Wikipedia documents. For the domain of mathematics, we opt for AutoMathText (Zhang et al., 2024), a carefully curated corpus derived from various sources including websites, arXiv, and GitHub. Each sample in this corpus has been labeled with a quality score from 0.0 ("the poorest") to 1.0 ("the best"), reflecting its relevance, quality, and educational value in the context of mathematical intelligence. Following previous work (Zhou et al., 2024), we specifically select those samples with scores higher than 0.7, containing about 0.7B tokens. For the field of code, we select the StarCoder (Li et al., 2023) corpus, which is widely recognized and employed in several studies (Luo et al., 2023b). It contains 86 programming languages, and we select the Python subset with approximately 1B tokens.

### A.2 GENERAL INSTRUCTION DATASETS

For general instruction datasets, we choose TULU-V2-mix (Ivison et al., 2023) and UltraFeedback (Cui et al., 2023) for instruction tuning and alignment, respectively. Specifically, each sample in TULU-V2-mix is either manually curated for quality or generated from GPT models to encourage complexity and diversity. We utilize the entire dataset of TULU-V2-mix (about 326K samples) mixed with domain-specific corpus for knowledge mixture continual pre-training (Section 2.1), and then select easy samples with the top-$K_1$ lowest perplexity score for subsequent instruction tuning. In addition, UltraFeedback is a widely-used diverse human preference alignment dataset, containing approximately 64K preference pairs. Similarly, we employ the whole dataset of UltraFeedback for knowledge mixture continual pre-training and then downsample easy pairs with the top-$K_2$ lowest perplexity score for alignment. It is noted that our TULU-V2-mix and UltraFeedback datasets are open-source and widely used in previous work (Meng et al., 2024; Hu et al., 2024a), ensuring a high level of transparency and facilitating fair experimental comparisons.

### A.3 BASELINES

In our experiments, we employ two prominent LLMs as the base model, *i.e., QWen2-7B* (Yang et al., 2024) and Meta-Llama-3-8B (Dubey et al., 2024). For comparative analysis, we consider the following three types of baseline methods:

• *Official Chat LLMs* consist of the official Chat LLMs that have undergone both instruction tuning and preference alignment using closed-source data. Here, we select two official chat LLMs corresponding to our selected base models, including *QWen2-7B-Chat* and *Llama-3-8B-Instruct*.

• *Reimplemented Chat LLMs* are developed by us following the processes of instruction tuning and alignment. Based on our selected three base LLMs, we conduct supervised fine-tuning (SFT) using TULU-V2-mix, followed by direct preference optimization (DPO) with UltraFeedback.

• *Continual Pre-training Augmented LLMs* include domain knowledge-enhanced Chat LLMs which initially undergo continual pre-training (CPT) with our domain-specific corpus, followed by the same implementation of supervised fine-tuning (SFT) and direct preference optimization (DPO) using TULU-V2-mix and UltraFeedback as open-source chat LLMs.

---

[2]https://dumps.wikimedia.org/enwiki/20240301/

Table 4: The categorization of the evaluation benchmarks

| | | |
|---|---|---|
| Specialized Domain | Wiki | NQ, TQ |
| | Math | GSM8K, MATH |
| | Code | MBPP, HumanEval |
| General Domain | Reading Comprehension | RACE-Hard, OpenBookQA |
| | Complex Reasoning | HellaSwag, CSQA, PIQA |
| | Examination | MMLU, BBH, ARC-Challenge, C-EVAL |
| | Instruction Following | MT-Bench |

## A.4 EVALUATION BENCHMARKS AND METRICS

For a comprehensive evaluation, we evaluate seven distinctive capabilities of LLMs based on a total of 16 representative NLP datasets:

• *Factual Question Answering* assesses the factual knowledge of LLMs in the Wikipedia domain. We employ NaturalQuestion (NQ) (Kwiatkowski et al., 2019) and TrivialQA (TQ) (Joshi et al., 2017) datasets and use the *Exact Match (EM)* metric to determine if the prediction is the same as the gold answer.

• *Math Reasoning* tests the LLMs' ability to solve mathematical problems. We use the GSM8K (Cobbe et al., 2021) and MATH (Hendrycks et al., 2021b) datasets, and evaluate predictions using the *Accuracy* metric.

• *Code Reasoning* tests the LLMs' ability to solve programming problems. We use MBPP (Austin et al., 2021) and HumanEval (Chen et al., 2021) datasets, with the *Pass@K* metric assessing the likelihood that at least one of the top-$K$ generated code samples for a problem passes the unit tests.

• *Reading Comprehension* measures the LLMs' ability to comprehend a passage and answer related questions. We use the RACE-Hard (Lai et al., 2017) and OpenBookQA (Mihaylov et al., 2018) datasets, and employ the *Exact Match (EM)* metric.

• *Commonsense Reasoning* evaluates the ability to answer questions using commonsense knowledge. We use the HellaSwag (Zellers et al., 2019), CSQA (Talmor et al., 2019), and PIQA (Bisk et al., 2020) datasets, and employ the *Accuracy* metric.

• *Examination* includes comprehensive and challenging benchmarks designed to assess problem-solving ability across various domains. We use MMLU (Hendrycks et al., 2021a), BBH (Suzgun et al., 2023), and ARC-Challenge (Bhakthavatsalam et al., 2021) for English examinations and C-EVAL (Huang et al., 2023) for Chinese. Both benchmarks are evaluated using the *Accuracy* metric.

• *Instruction Following* assesses the LLMs' ability to engage in coherent, informative, and engaging conversations. We use the MT-Bench (Zheng et al., 2023) datasets. For evaluation, we utilize the GPT-4-turbo [3] as the judging model following the OpenCompass settings, assigning a score ranging from 1 to 10 to the answer. We multiply this score by ten, resulting in a final score of 100.

Specifically, we evaluate the above datasets based on the OpenCompass framework (Contributors, 2023), which is a one-stop platform for large model evaluation, aiming to provide a fair, open, and reproducible benchmark for large model evaluation. We summarize the datasets in Table 4

## A.5 IMPLEMENTATION DETAILS

For both models, we conduct the mixed continual pre-training with the same hyper-parameters. Specifically, we set the batch size as 1920, the maximal sequence length as 2048, the maximal and minimal learning rate as 2e-5 and 5e-6, and the co-efficient $\alpha$ as 0.8. Then, for LLaMA3-8B model, we select the top 10,000 samples and 10,000 samples for SFT and DPO, respectively. For the SFT

---

[3] https://platform.openai.com/docs/models

stage, we perform 2 epochs of training with a learning rate of 5e-6 and a batch size of 128. For the DPO stage, we perform 2 epochs of training with a learning rate of 5e-7, a batch size of 128, and a $\beta$ of 0.01. For the QWen2-7B model, we select the top 10,000 samples and 10,000 samples for SFT and DPO, respectively. For the SFT stage, we perform 2 epochs of training with a learning rate of 5e-7 and a batch size of 128. For the DPO stage, we perform 2 epochs of training with a learning rate of 1e-7, a batch size of 128, and a $\beta$ of 0.1. When evaluating, we set the temperature as 0.3 and top-p as 0.9 for MT-Bench according to the official guideline and set the temperature as 0 for other benchmarks to control the randomness. As for the computational cost, the additional overhead introduced by the instruction and alignment data may be negligible, which only introduces approximately 0.09B tokens in continual pre-training and 0.008B tokens for totally selected 20K pairs in the format alignment stage.

