# OpenReview forum: "Mix-CPT: A Domain Adaptation Framework via Decoupling Knowledge Learning and Format Alignment"
_ICLR.cc/2025/Conference — ICLR 2025 Poster_

### Official Review · Reviewer_Yvdt · 2024-10-24

**Soundness:** 2
**Presentation:** 3
**Contribution:** 2
**Rating:** 3
**Confidence:** 5

**Summary:**

This paper proposes the Mix-CPT strategy to mix knowledge memorization and utilization into the continual pre-training stage. Specifically, the authors eliminate the chat template of general instruction-following and preference-alignment data (which is traditionally used in the SFT stage with chat templates), and mix them with domain knowledge corpus for vanilla next-token prediction optimization. A logit swap self-distillation strategy is developed to mitigate catastrophic forgetting. After that, the authors derived a format alignment score to select a small number of instructions from the CPT dataset to execute SFT and DPO, in order to focus mainly on pure style or format learning for downstream tasks.

**Strengths:**

1. The research point is important. Efficient knowledge memorization is crucial to derive both fundamental and domain-specific LLMs.
2. The idea of decoupling knowledge learning and format alignment has promising efficacy, as they serve different purposes.
3. The paper is well-written and easy to follow.

**Weaknesses:**

1. My core concern is the lack of novelty. Taking SFT samples into the CPT stage has been explored by several works[1][2], but this paper neither discusses the methodology difference nor compares the experimental performance. Besides, some influential technical reports[3] have pointed out that moving SFT samples to CPT cannot help the final model performance, since they will eventually go through the SFT and preference alignment stages. I wonder how the authors consider this problem.
2. I'm not convinced whether general alignment training samples can truly teach LLMs to utilize domain knowledge to answer professional questions. In lines 73-75, the referred two papers (RLHF and DPO) cannot support this hypothesis.
3. The experiment setting is unreasonable. Math, code, and wiki are hard to be considered as 'domain knowledge', since they have already been exposed to LLM's pre-training corpus and take a considerable proportion.  The 'knowledge injection' should involve specific domains[1][2] like medicine, finance, etc.

[1] Adapting large language models via reading comprehension. 2023.

[2] Instruction pre-training: Language models are supervised multitask learners, 2024.

[3] Deepseek llm: Scaling open-source language models with longtermism. 2024.

**Questions:**

1. What is the FAS score's physical meaning?  Does a higher score mean a better or worse sample? Besides, if the authors aim at selecting the samples that LLM have not currently learned the format, taking Eq.8 for an example, it seems more reasonable to compare the loss on response (r) conditioned on questions with and without chat templates, i.e., $\frac{L(r|\hat{q})}{L(r|\bar{q})}$. And Eq.9 presents the same problem.

---

> ### Author Response · Authors · 2024-11-24
> **Response to the Concerns of Reviewer [Part 1]**
>
> Thank you for patiently awaiting our response, since we have conducted extensive experiments to address your concerns. Next are the details.
>
> **1. My core concern is the lack of novelty. Taking SFT samples into the CPT stage has been explored by several works[1][2], but this paper neither discusses the methodology difference nor compares the experimental performance. Besides, some influential technical reports[3] have pointed out that moving SFT samples to CPT cannot help the final model performance, since they will eventually go through the SFT and preference alignment stages. I wonder how the authors consider this problem.**
>
> Thanks for your comment. We respond to your three concerns as follows:
>
> For your first concern: In the second paragraph of Introduction (line 49-53), we have discussed a lot about the limitations of previous domain adaptation methods. Our Mix-CPT paradigm stands out from previous work in two key aspects:
> - First, we noticed that the two goals of knowledge utilization and format alignment are divergent in model optimization, and knowledge memorization can benefit from knowledge utilization by learning to elicit the acquired knowledge. Therefore, we propose a new domain adaptation framework that includes **domain knowledge learning and general format alignment** stages.
> - Second, we noticed that existing approaches still **suffer from the catastrophic forgetting issue** in the general domain seriously while adapting to specific domains. Therefore, we propose the **Logit Swap Self-Distillation (LSSD)** technique to maintain the general capabilities.
>
> For your second concern: The method [1][2] focuses **solely on enhancing the domain-specific capabilities** of LLMs and relies on domain models to construct domain-related instructional data, **without exploring the decoupling of instruction learning and the maintaining of the model's general capabilities**. So we finally do not compare to the methods in our paper. According to your suggestions, following the experimental setup of Instruction Pretraining, we supplement experiments on the LLaMA-3-8B-Base model in the **medical domain** and compared the performance between their method and ours. The experimental results are shown in the below table.
>
>  **Method** | **USMLE** | **RCT** | **PubMedQA** | **MQP** | **Domain Average** | **MATH** | **MBPP** | **MMLU** | **CEVAL** | **General Average**
> ---|----|---|----|----|----|----|----|----|-----|------
>  **LLaMA3-8B-Base**|40.6|73.6|59.8|66.2|60.1|18.0|57.9|65.9|49.3|**47.8**
>  **domain_raw+domain_instruction+CPT**|40.1|70.6|56.7|55.4|55.7|5.9|57.0|63.2|48.6|43.7
>  **domain_raw+domain_instruction+Mix_CPT**|40.0|73.4|61.0|55.7|56.4|7.9|57.5|64.0|50.4|45.0
>  **domain_raw+general_instruction+Mix_CPT (Ours)**|40.5|71.0|63.4|79.2|**63.4**|16.6|58.0|66.2|49.7|47.6
>
> We can see that:
> 1. Direct conducting CPT with domain raw data and instructions, constrained by issues such as catastrophic forgetting, results in a significant performance decline across all domains. The original paper further integrated an equivalent amount of general instruction data to implement the performance improvement. In contrast, using the same domain raw data and instructions, our Mix-CPT method based on the self-distillation technique can learn the domain-specific capabilities to some extent, while better maintaining its general capabilities.
> 2. Furthermore, when employing domain raw data alongside general instructions and applying the self-distillation technique to mitigate catastrophic forgetting, the model exhibits improvements in both domain-specific and general capabilities. This demonstrates that our proposed method of general knowledge utilization capabilities can be successfully transferred to the utilization of domain-specific knowledge.
>
> For your third concern, we think that the report [3] also acknowledges mixing instruction data with pre-training data. We consider that their failure in mixing the instruction and pre-training data might stem from two aspects:
> 1. They **only use the traditional language modeling loss** for the mixture data. In Table 1 of our paper, the results have verified that CPT with only language modeling loss results in the issue of catastrophic forgetting where the domain performance increases but the general performance decreases. This motivates us to propose a logit swap self-distillation loss combined with the language modeling loss to alleviate this issue.
> 2. They **only employ multi-choice questions to mix with the pre-training data**. This kind of instruction is relatively simplistic, which can constrain the performance and generalizability of domain-specific knowledge in the final model. So mixing domain corpus and general alignment data will enforce knowledge memorization and utilization but require to ensure the diversity of instructions and avoid catastrophic forgetting.

---

> ### Author Response · Authors · 2024-11-24
> **Response to the Concerns of Reviewer [Part 2]**
>
> **2. I'm not convinced whether general alignment training samples can truly teach LLMs to utilize domain knowledge to answer professional questions. In lines 73-75, the referred two papers (RLHF and DPO) cannot support this hypothesis.**
>
> Thanks for your comment. We sincerely apologize for the unintentional mistake in the reference. Actually, repurposing general instruction data is inspired by two works [4][5]:
> 1. The work [4] points out that **it is beneficial to deliberately expose LLMs to QA data** so that the process of memorizing knowledge can take into account how this knowledge is accessed through questions. However, this work relies on proprietary models to synthesize high-quality domain-specific instructions, which is not practicable in real-world scenarios.
> 2. On the other side, the work [5] discovers that **base LLMs and their alignment-tuned versions perform nearly identically in decoding** on the majority of token positions and most distribution shifts occur with stylistic tokens (e.g., discourse markers, safety disclaimers).
>
> This direct evidence strongly supports the hypothesis that the knowledge utilization capability is not related to specific domain instructions and alignment tuning primarily learns to adopt the language style of AI assistants. So we repurpose the general instruction data to teach LLMs to utilize domain knowledge. We will revise our references to make our paper clearer. Thanks for pointing out this issue again.
>
> Furthermore, we conduct experiments on the **medical domain** with the same number of domain instructions and generation instructions, respectively. We show the results in the following Table. **We find that when given the same number of general instructions and domain-specific instructions, the general instructions better facilitate the learning of domain-specific capabilities while maintaining general competencies**. This experimental result aligns with the conclusions of the aforementioned paper and also demonstrates our claims.
>
> **Method** | **USMLE** | **RCT** | **PubMedQA** | **MQP** | **Domain Average** | **MATH** | **MBPP** | **MMLU** | **CEVAL** | **General Average**
> ---|----|---|----|----|----|----|----|----|-----|------
>  **LLaMA3-8B-Base**|40.6|73.6|59.8|66.2|60.1|18.0|57.9|65.9|49.3|**47.8**
>  **domain_raw+domain_instruction+Mix_CPT**|40.0|73.4|61.0|55.7|56.4|7.9|57.5|64.0|50.4|45.0
>  **domain_raw+general_instruction+Mix_CPT (Ours)**|40.5|71.0|63.4|79.2|**63.4**|16.6|58.0|66.2|49.7|47.6
>
> ————————————————————————————————————————————————————————————
>
> **3. The experiment setting is unreasonable. Math, code, and wiki are hard to be considered as 'domain knowledge', since they have already been exposed to LLM's pre-training corpus and take a considerable proportion. The 'knowledge injection' should involve specific domains[1][2] like medicine, finance, etc.**
>
> We appreciate the concern that math, code, and Wikipedia may not represent traditional "domain-specific" knowledge, but we selected these domains to evaluate our framework under controlled conditions where the domain knowledge is both measurable and representative of specific tasks.
> 1. First, math, code, and Wikipedia provide standardized datasets with clear benchmarks for evaluating domain adaptation methods. These domains allow for reproducible experiments and comparisons against prior work, enabling us to validate the efficacy of the Mix-CPT framework in improving domain task performance.
> 2. Second, by focusing on these domains, we can analyze the impact of knowledge injection in scenarios where the model already has partial exposure. This controlled setup provides insights into how well Mix-CPT balances knowledge retention and adaptation.
>
> To address your concerns, we extend our method to more specialized domains such as the **medicine domain** following existing work [1]. We can see that, based on the identical domain raw data and general instructions, we observe that compared to traditional CPT methods, our proposed Mix-CPT approach more effectively facilitates the learning of domain-specific skills and better maintain the general capabilities, indicating the effectiveness and generalizability of our method.
>
> **Method** | **USMLE** | **RCT** | **PubMedQA** | **MQP** | **Domain Average** | **MATH** | **MBPP** | **MMLU** | **CEVAL** | **General Average**
> ---|----|---|----|----|----|----|----|----|-----|------
>  **LLaMA3-8B-Base**|40.6|73.6|59.8|66.2|60.1|18.0|57.9|65.9|49.3|**47.8**
>  **CPT**|35.5|72.4|65.1|76.4|62.3|14.5|56.9|64.2|46.6|45.6
>  **Mix_CPT (Ours)**|40.5|71.0|63.4|79.2|**63.4**|16.6|58.0|66.2|49.7|47.6

---

> ### Author Response · Authors · 2024-11-24
> **Response to the Concerns of Reviewer [Part 3]**
>
> **4. What is the FAS score's physical meaning?  Does a higher score mean a better or worse sample? Besides, if the authors aim at selecting the samples that LLM have not currently learned the format, taking Eq.8 for an example, it seems more reasonable to compare the loss on response (r) conditioned on questions with and without chat templates, i.e., L(r|q^)L(r|q¯). And Eq.9 presents the same problem.**
>
> We thank the reviewer for raising these insightful questions regarding the FAS score and its role in our framework.
>
> **For the physical meaning of the FAS Score**, it represents **the difficulty of the model to handle response with formatted instruction**. Specifically:
> - Higher FAS Scores: Indicate that when given formatted instructions, the difficulty for the model to handle responses is significantly large. This suggests that although the model possesses the response knowledge, it struggles to utilize this knowledge while following a formatted instruction.
> - Lower FAS Scores: Indicate that the formatted instructions do not affect the model’s ability to utilize existing knowledge. This suggests that when the model is provided with formatted instructions, the difficulty in modeling responses does not change.
> Thus, higher scores are prioritized during sample selection to target areas where the model has room for improvement.
>
> **For the formulation of the FAS Score**, we consider that the objective of the instruction tuning stage is to facilitate the instruction following the ability of LLMs. So Eq. 8 can measure the difficulty of a response the LLM can handle when given formatted instructions, as denoted by L(r|q^)/L(r). Similarly, the objective of preference alignment is to distinguish between the positive response and negative response when given a formatted question. So the FAS score for alignment data is formulated as L(r+|q^)/L(r-/q^).
>
> According to your suggestions, we also conduct experiments by using your proposed formulation. We evaluate the LLaMA-3-8B-Base model on Code, Math, Examination, and Instruction Following using the evaluation setting in Table 2. We show the results in the following Table. We observe that our proposed method performs slightly below the variant method on the instruction-following evaluation set. However, it achieves superior performance on other evaluation sets, ultimately resulting in better average performance. This demonstrates the effectiveness of the FAS Score.
>
> **Method** | **Code** | **Math** | **Examination** | **Instruction Following** | **Average**
> ---|----|---|----|----|----
>  **L(r\|q^)/L(r\|q)**|51.0|38.0|59.1|73.2|55.3
>  **L(r\|q^)/L(r) (Ours)**|52.7|38.3|60.7|72.1| **56.0**
>
>
>
> —————————————————————————————————————————————————————————
>
> **Reference**
>
> [1] Adapting large language models via reading comprehension. 2023.
>
> [2] Instruction pre-training: Language models are supervised multitask learners, 2024.
>
> [3] Deepseek llm: Scaling open-source language models with longtermism. 2024.
>
> [4] Instruction-tuned Language Models are Better Knowledge Learners, 2024.
>
> [5] The Unlocking Spell on Base LLMs: Rethinking Alignment via In-Context Learning, 2024.

---

> > ### Comment · Reviewer_Yvdt · 2024-11-26
> >
> > I appreciate the author's responses, which however have not yet resolved my concerns.
> >
> > Let's consider a specific scenario: if a LLM is required to make a medical diagnosis based on the descriptions of a patient, how it is possible to learn to relate the patient's descriptions to corresponding medical knowledge points and make the final diagnosis, after training with general SFT data only? And the model trained with MixCPT will eventually go through post-training (SFT and preference alignment), where LLMs are further trained to follow instructions and human preferences far beyond what MixCPT can achieve. I'm still confused about the role MixCPT plays in the entire paradigm. Besides, the evaluation protocol and experimental results are also questionable. It is unclear whether models are evaluated in a few-shot or zero-shot manner, and why the performance on MQP varies so rapidly.

---

> > > ### Author Response · Authors · 2024-11-27
> > > **Response to the Concerns of Reviewer [Part 2]**
> > >
> > > **2. For you second concern about the role of MixCPT**
> > >
> > > We would like to clarify that formatting the SFT and DPO data in our first mixture pre-training stage is simply removing any chat templates and directly concatenating the instruction and response parts. This manner aims to convert the SFT and DPO data into a sequence like the raw domain document. However, the model after mixture pre-training is just a base model, so we still need additional alignment stages (adding the chat template to the SFT and DPO data) to achieve the format alignment objective for obtaining a chat model that can interact with humans. These selected alignment samples have been encountered during the mixture pre-training stage, so we only select a small number of samples and add the chat template to them. Therefore, the first mixture pre-training stage is important to memorize the domain knowledge and learn how to utilize it. And the following SFT and DPO stages just aim to convert the base model to a chat model that is able to respond with formatted templates.
> > >
> > > **3. For your third concern about experimental settings**
> > >
> > > The models are evaluated in a few-shot manner following existing work [1]. Because MQP consists of Yes/No questions and only contains very few samples (about 600 samples), the performance of MQP varies so rapidly compared to other large-scale multiple-choice datasets.
> > >
> > > [1] Instruction pre-training: Language models are supervised multitask learners, 2024.

---

> ### Author Response · Authors · 2024-11-27
> **Response to the Concerns of Reviewer [Part 1]**
>
> Thank you for your response! I would like to further explain your three concerns.
>
> **1. For your first concern about the role of general SFT data**
>
> We try to explain that through an example and then give some further experiments.
>
> ----
>
> **Medical Domain Corpus**: Fever and sore throat are common symptoms of viral or bacterial infections. Streptococcal pharyngitis is a bacterial cause, often accompanied by swollen lymph nodes and the absence of cough.
>
> **General SFT Data**:
> - Instruction: What is an animal with black and white stripes on its body?
> - Response: An animal with black and white stripes on its body is a zebra. These stripes are unique to each individual and serve purposes such as camouflage, temperature regulation, and social signaling.
>
> We think that the SFT data teachs the model how to relate the characteristics to a target entity or knowledge. When we apply the model in the medical instruction below, it can learn how to relate the symptoms to a disease.
>
> **Medical instruction**: What's wrong with me? I have had a fever and sore throat for 3 days. I also feel tired but haven’t been coughing.
>
> **Response**: Based on your symptoms, it’s possible you have streptococcal pharyngitis, a bacterial throat infection. This condition is often associated with fever, sore throat, swollen lymph nodes, and an absence of cough.
>
> ----
>
> The above is a conceptual example to explain the role of general SFT data in the utilization of domain knowledge. We understand that the impact of general SFT data in utilizing domain knowledge is not as direct as that of domain-specific SFT data. However, what we aim to emphasize in this paper is that **when domain-specific SFT data is scarce or limited in scale**, we can effectively leverage widely accessible general SFT data to enhance performance on domain-specific tasks. **If the domain SFT data is available, it will be better to mix the general and domain SFT data**.
>
> **We conduct an experiment comparing only the use of domain or general SFT data to mixing both kinds of data as below.** We can see that:
> 1. Using only domain-specific instructions to perform CPT significantly impacts the overall model performance. It is essential to blend general instructions to acquire domain capabilities effectively. However, this approach leads to the problem of knowledge forgetting, resulting in a noticeable decline in general performance (47.8 --> 43.7), aligning with the existing results [1].
> 2. Our method demonstrates that using only general instructions allows the model to acquire domain-specific capabilities to a certain extent while maintaining general abilities, demonstrating the importance of general instructions.
> 3. Furthermore, by incorporating domain-specific instructions, we can achieve additional improvements in domain tasks (63.4 --> 64.4). Concurrently, compared to traditional CPT methods, our approach better preserves original performance on general tasks (45.3 --> 47.0).
>
> In conclusion, our contributions are first to demonstrate that general SFT data can also significantly benefit domain adaptation when domain-specific SFT data is scarce or limited in scale, and mixing both domain and general instructions can be more helpful. Moreover, our proposed **logit swap self-distillation can maintain the general and domain performance to achieve a good balance**.
>
>  **Method** | **USMLE** | **RCT** | **PubMedQA** | **MQP** | **Domain Average** | **MATH** | **MBPP** | **MMLU** | **CEVAL** | **General Average**
> ---|----|---|----|----|----|----|----|----|-----|------
>  **LLaMA3-8B-Base**|40.6|73.6|59.8|66.2|60.1|18.0|57.9|65.9|49.3|**47.8**
>  **domain_raw+domain_instruction+CPT**|40.1|70.6|56.7|55.4|55.7|5.9|57.0|63.2|48.6|43.7
>  **domain_raw+domain_instruction+general_instruction+CPT**|36.1|74.4|70.1|78.4|**64.8**|14.2|55.8|63.6|47.7|45.3
>  **domain_raw+general_instruction+Mix_CPT (Ours)**|40.5|71.0|63.4|79.2|63.4|16.6|58.0|66.2|49.7|47.6
>  **domain_raw+domain_instruction+general_instruction+Mix_CPT (Ours)**|35.5|73.0|71.0|78.2|64.4|16.6|56.5|66.0|48.9|47.0

---

> ### Author Response · Authors · 2024-11-29
> **Kindly Reminder for the  Further Discussion**
>
> Dear Yvdt,
>
> Thanks for your valuable comments. We try our best to elaborate on your concerns and add extra experiments to demonstrate our claims. We would like to know whether you find our response satisfactory, or if there are more questions that we could clarify. Since the rebuttal stage is coming to an end, we are more than happy to hear your comments and address any of your further concerns during the remaining time.
>
> Thanks,
>
> Authors

---

> ### Author Response · Authors · 2024-11-30
> **A Friendly Reminder for Discussion**
>
> Dear Yvdt,
>
> Thanks for your valuable comments. We try our best to **elaborate on your concerns and add extra experiments** to demonstrate our claims. We would like to know **whether you find our response satisfactory, or if there are more questions that we could clarify**. Since the rebuttal stage is coming to an end, we are more than happy to hear your comments and address any of your further concerns during the remaining time.
>
> Thanks,
>
> Authors

---

> > ### Author Response · Authors · 2024-12-01
> > **Kindly Reminder for the Further Discussion**
> >
> > Dear Yvdt,
> >
> > Thanks for your valuable comments. We try our best to **elaborate on your concerns and add extra experiments to demonstrate our claims**. We would like to know whether you find our response satisfactory, or if there are more questions that we could clarify. Since the rebuttal stage is coming to an end, we are more than happy to hear your comments and address any of your further concerns during the remaining time.
> >
> > Thanks,
> >
> > Authors

---

> > > ### Comment · Reviewer_Yvdt · 2024-12-02
> > >
> > > Thanks for your response. In the past discussion, the reason why I doubted the experimental results had been confirmed, that is, the few-shot evaluation of base models after MixCPT in the medical corpus can be biased, as the few-shot medical instruction examples can teach them to answer medical questions. The authors claim that they only need general instruction/alignment data to adapt base LLMs to downstream domains, so they should test the chat model on real-world medical QA data. For example, the authors can continually pre-train a base model on MMedC[1], post-train the model on general SFT and alignment data, and evaluate the derived chat model on MMedBench[1] in a zero-shot manner. A more rigorous setting like this can strongly verify the authors' claims.
> > >
> > > [1] Qiu P, Wu C, Zhang X, et al. Towards building multilingual language model for medicine[J]. Nature Communications, 2024, 15(1): 8384.

---

### Official Review · Reviewer_wQCa · 2024-11-03

**Soundness:** 3
**Presentation:** 4
**Contribution:** 3
**Rating:** 6
**Confidence:** 3

**Summary:**

The paper introduces Mix-CPT, a domain adaptation framework for LLMs that separates knowledge learning from format alignment to enhance efficiency and prevent knowledge loss. In the first stage, Mix-CPT trains on a mixture of domain-specific and general instructional data to balance domain learning and general capabilities, using Logit Swap Self-Distillation to retain prior knowledge. The second stage, format alignment, fine-tunes the model with a few selected samples to align responses to human-preferred formats. Extensive experiments and ablations have been carried out to support the proposed framework.

**Strengths:**

1. The paper introduces a novel view of CPT that separates domain knowledge learning from format alignment, potentially enhances both domain adaptation efficiency and task-specific performance.

2. The proposed LSSD method to avoid catastrophic forgetting is clear and straightforward, allowing the model to adapt without sacrificing previously learned information.

3. Integrating domain-specific raw data with general instructional data during the pre-training phase is a strength, as it appears to balance both domain-specific and general knowledge effectively.

**Weaknesses:**

see questions.

**Questions:**

1. The results indicate a dependence on the quality of domain-specific data, with lower-quality data adversely impacting performance (Fig. 3). This reliance could restrict the framework’s applicability in real-world scenarios where high-quality datasets are limited.

2. Why is the logit of the top-1 token specifically swapped with the ground truth in LSSD? Beyond the math and average measurements in Fig. 6, could you provide a more detailed explanation of why this technique is effective?

3. Could you provide an analysis of computational efficiency compared to traditional domain adaptation frameworks?

---

> ### Author Response · Authors · 2024-11-24
> **Response to the Concerns of Reviewer [Part 1]**
>
> Thank you for patiently awaiting our response, since we have conducted extensive experiments to address your concerns. Next are the details.
>
> **1. The results indicate a dependence on the quality of domain-specific data, with lower-quality data adversely impacting performance (Fig. 3). This reliance could restrict the framework’s applicability in real-world scenarios where high-quality datasets are limited.**
>
> Thanks for your comments. **We instead believe that the conclusions drawn from Figure 3 will strongly support the applicability of our method in real-world scenarios where the domain data is scarce**. In such domains, obtaining large-scale pre-training data is often infeasible. However, by employing techniques such as data synthesis, knowledge distillation, or multilingual translation, combined with appropriate data filtering and quality evaluation, it is possible to obtain a small amount of high-quality data. This approach is likely to yield better results than relying on large-scale but low-quality data for pre-training. But in reality, it will be comprehensive to consider both the scale and quality of pre-training data to achieve optimal performance.
>
> **2. Why is the logit of the top-1 token specifically swapped with the ground truth in LSSD? Beyond the math and average measurements in Fig. 6, could you provide a more detailed explanation of why this technique is effective?**
>
> The Logit Swap Self-Distillation (LSSD) mechanism is designed to balance two critical goals during continual pre-training: **Retain Previously Learned Knowledge** and **Adapt to Domain-Specific Knowledge**. The choice to specifically swap the logit of the top-1 predicted token with that of the ground truth token is motivated by the following considerations:
> 1. **Targeted Adjustment of Predictions**: Swapping the logit of the top-1 predicted token focuses the adjustment on the token that the model is most confident about. This ensures that the model’s most probable prediction is corrected toward the ground truth without unnecessarily disrupting the rest of the prediction distribution.
> 2. **Preservation of Knowledge Distribution**: Unlike traditional distillation methods that modify all logits based on the ground truth, LSSD selectively intervenes on the logit of the top-1 token. This preserves the probabilities associated with other tokens, reducing the risk of overwriting general knowledge that the model has already acquired.
> 3. **Smooth Transition to New Knowledge**: The logit swap introduces a subtle but effective adjustment by aligning the ground truth token’s probability with the model’s original predictions. The approach respects the base model’s pre-existing strengths and gradually adapts it to new data, leading to more stable learning outcomes.

---

> ### Author Response · Authors · 2024-11-24
> **Response to the Concerns of Reviewer [Part 2]**
>
> **3. Could you provide an analysis of computational efficiency compared to traditional domain adaptation frameworks?**
>
> Thanks for your comment. Compared to traditional domain adaptation frameworks, we mainly incorporate additional SFT and DPO data in the CPT stage and propose a logit swap self-distillation loss, which requires the base model to compute the original logits.
> 1. In Section 3.1 (lines 292-296), we have discussed that the **additional overhead introduced by the instruction and alignment data may be negligible**, which only introduces approximately **0.09B** tokens in continual pre-training and 0.008B tokens for total selected 20K pairs in the format alignment stage.
> 2. For the LSSD loss, in fact, since the self-distillation model is the base LLM, **it can be distilled either online or offline**. Specifically, if faster processing time is desired, offline distillation can be used by saving the word logit distribution of the data and then loading the data during training. However, this may require substantial hard drive storage. If storage conservation is preferred, online distillation can be utilized, where each batch of data is first inferred using the base LLM. The cost of this is significantly lower than updating the model based on the batch data, which also includes backpropagation updates.
>
> Inspired by your concern, we also conduct additional experiments by only **randomly employing 50% pre-training data for performing the logit swap self-distillation**, which can reduce half of the computational costs introduced by self-distillation. We conduct the experiment on the medical domain with the LLaMA3-8B-Base model and evaluate its performance on the domain-specific tasks and general tasks as described in Section 3.1. We show the results in the following Table. We find that, compared to distilling the entire dataset, reducing the distillation data by half further enhances the model's performance on domain-specific tasks, with only a slight decrease in general capabilities. This suggests that exploring even lower distillation ratios in the future could achieve a greater tradeoff between computation cost and performance. Due to constraints in computational resources, we plan to include more different distillation ratios in the future version.
>
> **Method** | **USMLE** | **RCT** | **PubMedQA** | **MQP** | **Domain Average** | **MATH** | **MBPP** | **MMLU** | **CEVAL** | **General Average**
> ---|----|---|----|----|----|----|----|----|-----|------
>  **LLaMA3-8B-Base**|40.6|73.6|59.8|66.2|60.1|18.0|57.9|65.9|49.3|**47.8**
>  **+Mix_CPT**|40.5|71.0|63.4|79.2|63.4|16.6|58.0|66.2|49.7|47.6
>  **+Mix_CPT (50% LSSD)**|40.3|71.2|64.1|83.0|**64.7**|16.0|57.7|65.5|50.3|47.4

---

> > ### Comment · Reviewer_wQCa · 2024-11-27
> >
> > Thank you for your response. My concerns have been addressed, I will keep my rating.

---

> > > ### Author Response · Authors · 2024-11-29
> > > **Response to Reviewer**
> > >
> > > Dear wQCa,
> > >
> > > Thank you for your valuable comments. **We sincerely thank you for the positive reply and the continued vote to accept the work!**. We are ready to address any of your further concerns during the remaining time.

---

### Official Review · Reviewer_9g2u · 2024-11-04

**Soundness:** 3
**Presentation:** 3
**Contribution:** 2
**Rating:** 6
**Confidence:** 3

**Summary:**

The paper proposes a novel two-stage domain adaptation method for language models. The authors discuss the limitations of the existing training pipeline, which includes pre-training, instruction-tuning, and preference alignment. Based on this discussion, they propose a mix of data from each stage in an appropriate format to enable the model to learn both domain knowledge (i.e., pre-training) and knowledge utilization (i.e., instruction-tuning and preference learning). Additionally, the authors enhance the approach by training the model further on a small yet essential subset of data used in pre-training for improved model output alignment. The proposed method was evaluated with QWen2-7B and Llama-3-8B on domain data, including Wiki data, AutoMathText, and StarCoder. This method outperforms the simple CPT model.

**Strengths:**

- The proposed method is well-motivated
- The paper is well-written and easy to follow
- The proposed method outperforms the baselines

**Weaknesses:**

- Logit swap self-distillation requires the base LLM to perform inferences on the entire training dataset. This process can be quite expensive, especially with large datasets and models.
- A concern with the experimental setup is that Wikipedia data is used for domain adaptation. However, Wikipedia is more general-purpose than domain-specific, as it covers a wide range of topics. Additionally, the base models have likely already been trained on this widely used open-source dataset, which weakens the observations drawn from the experiments.
- The authors should have compared their method to a recent, similar CPT method [1]. Method [1] has an advantage as it trains a base model on both raw documents and formatted reading comprehension data. This approach is more straightforward than the authors’ method, as it requires only single-stage training and does not necessitate the separate collection of CPT, SFT, and DPO data.
- It is unclear why the model requires a second alignment stage, as it has already been trained on formatted DPO data. Is this due to issues with the initial stage of training?

[1] Adapting Large Language Models to Domains via Reading Comprehension, ICLR 2024

**Questions:**

- Refer to the weakness.

---

> ### Author Response · Authors · 2024-11-24
> **Response to the Concerns of Reviewer [Part 1]**
>
> Thank you for patiently awaiting our response, since we have conducted extensive experiments to address your concerns. Next are the details.
>
> **1. Logit swap self-distillation requires the base LLM to perform inferences on the entire training dataset. This process can be quite expensive, especially with large datasets and models.**
>
> Thanks for your comments. We understand your concern very well but this issue might be trivial in our framework compared to the pre-training cost. In fact, since the self-distillation model is the base LLM, **it can be distilled either online or offline**. Specifically, if faster processing time is desired, offline distillation with hardware speedup, such as vLLM, can be used by saving the word logit distribution of the data and then directly loading the data during training. However, this may require the hard drive storage. If storage conservation is preferred, online distillation can be utilized, where each batch of data is first inferred using the base LLM.
>
> Inspired by your concern, we also conduct additional experiments by only **randomly employing 50% pre-training data for performing the logit swap self-distillation**, which can reduce half of the computational costs introduced by self-distillation. We conduct the experiment on the medical domain (consider that your Question 2) with the LLaMA3-8B-Base model and evaluate its performance on the domain-specific tasks and general tasks as described in Section 3.1. We show the results in the following Table. We find that, compared to distilling the entire dataset, reducing the distillation data by half further enhances the model's performance on domain-specific tasks, with only a slight decrease in general capabilities. This suggests that exploring even lower distillation ratios in the future could achieve a greater tradeoff between computation cost and performance. Due to constraints in computational resources, we plan to include more different distillation ratios in the future version.
>
> **Method** | **USMLE** | **RCT** | **PubMedQA** | **MQP** | **Domain Average** | **MATH** | **MBPP** | **MMLU** | **CEVAL** | **General Average**
> ---|----|---|----|----|----|----|----|----|-----|------
>  **LLaMA3-8B-Base**|40.6|73.6|59.8|66.2|60.1|18.0|57.9|65.9|49.3|**47.8**
>  **+Mix_CPT**|40.5|71.0|63.4|79.2|63.4|16.6|58.0|66.2|49.7|47.6
>  **+Mix_CPT (50% LSSD)**|40.3|71.2|64.1|83.0|**64.7**|16.0|57.7|65.5|50.3|47.4
>
> ——————————————————————————————————————————
>
> **2. A concern with the experimental setup is that Wikipedia data is used for domain adaptation. However, Wikipedia is more general-purpose than domain-specific, as it covers a wide range of topics. Additionally, the base models have likely already been trained on this widely used open-source dataset, which weakens the observations drawn from the experiments.**
>
> Thank you for your thoughtful feedback regarding our experimental setup and the use of Wikipedia data for domain adaptation. We first explain why we utilize Wikipedia and then conduct experiments on medical domain data.
>
> Firstly, we acknowledge that Wikipedia belongs to the domain of encyclopedia knowledge, which covers a wide range of topics and is **a valuable resource for foundational knowledge adaptation**. Specifically, Wikipedia’s factual and well-organized content provides a robust baseline for evaluating our framework’s ability to integrate and utilize diverse knowledge sources. In our study, Wikipedia served as a preliminary dataset to test the efficacy of our domain adaptation method before applying it to more domain-specific corpora (i.e., medical , math, code).
>
> Second, we further conduct experiments in the **medical domain** to demonstrate the effectiveness of our method following existing work [1]. We compare our methods with it with the same evaluation protocol. We show the results in the following table. We can see that, based on the identical domain raw data and general instructions, we observe that compared to traditional CPT methods, **our proposed Mix-CPT approach more effectively facilitates the learning of domain-specific skills and better maintains the general capabilities**, indicating the effectiveness and generalizability of our method.
>
> **Method** | **USMLE** | **RCT** | **PubMedQA** | **MQP** | **Domain Average** | **MATH** | **MBPP** | **MMLU** | **CEVAL** | **General Average**
> ---|----|---|----|----|----|----|----|----|-----|------
>  **LLaMA3-8B-Base**|40.6|73.6|59.8|66.2|60.1|18.0|57.9|65.9|49.3|**47.8**
>  **CPT**|35.5|72.4|65.1|76.4|62.3|14.5|56.9|64.2|46.6|45.6
>  **Mix_CPT**|40.5|71.0|63.4|79.2|**63.4**|16.6|58.0|66.2|49.7|47.6

---

> ### Author Response · Authors · 2024-11-24
> **Response to the Concerns of Reviewer [Part 2]**
>
> **3. The authors should have compared their method to a recent, similar CPT method [1]. Method [1] has an advantage as it trains a base model on both raw documents and formatted reading comprehension data. This approach is more straightforward than the authors’ method, as it requires only single-stage training and does not necessitate the separate collection of CPT, SFT, and DPO data.**
>
> Thanks for your comment. The method [1] requires synthesizing domain-specific instructions related to reading comprehension, therefore it is not compatible with the math and code domains in our experiments. The math and code corpora contain massive symbols and code snippets, which are not appropriate for synthesizing reading comprehension instructions. So we finally do not compare to the method in our paper. To address your concern, we try to mimic their settings and reuse existing instruction data **in medical domain** following existing work [1][2]. The experimental results are shown in the below table.
>
>  **Method** | **USMLE** | **RCT** | **PubMedQA** | **MQP** | **Domain Average** | **MATH** | **MBPP** | **MMLU** | **CEVAL** | **General Average**
> ---|----|---|----|----|----|----|----|----|-----|------
>  **LLaMA3-8B-Base**|40.6|73.6|59.8|66.2|60.1|18.0|57.9|65.9|49.3|**47.8**
>  **domain_raw+domain_instruction+CPT**|40.1|70.6|56.7|55.4|55.7|5.9|57.0|63.2|48.6|43.7
>  **domain_raw+domain_instruction+Mix_CPT**|40.0|73.4|61.0|55.7|56.4|7.9|57.5|64.0|50.4|45.0
>  **domain_raw+general_instruction+Mix_CPT (Ours)**|40.5|71.0|63.4|79.2|**63.4**|16.6|58.0|66.2|49.7|47.6
>
> We can see that:
> 1. Direct conducting CPT with domain raw data and instructions, constrained by issues such as catastrophic forgetting, results in a significant performance decline across all domains. In contrast, using the same domain raw data and instructions, our Mix-CPT method based on the self-distillation technique can learn the domain-specific capabilities to some extent, while better maintaining its general capabilities.
> 2. Furthermore, when employing domain raw data alongside general instructions and applying the self-distillation technique to mitigate catastrophic forgetting, the model exhibits improvements in both domain-specific and general capabilities. This demonstrates that our proposed method of general knowledge utilization capabilities can be successfully transferred to the utilization of domain-specific knowledge.
>
> ————————————————————————————————————————————————
>
> **4. It is unclear why the model requires a second alignment stage, as it has already been trained on formatted DPO data. Is this due to issues with the initial stage of training?**
>
> Thanks for your comment. We would like to clarify that formatting the SFT and DPO data in our first mixture pre-training stage is simply removing any chat templates and directly concatenating the instruction and response parts. This manner aims to convert the SFT and DPO data into a sequence like the raw domain document. However, **the model after mixture pre-training is just a base model**, so we still need additional alignment stages (adding the chat template to the SFT and DPO data) to achieve the format alignment for obtaining a chat model that can interact with humans. This alignment stage can be implemented more efficiently with fewer samples compared to previous work [1][2]. We also present the results of base LLMs (without alignment stage) and chat LLMs in Tables 1 and 2, respectively.
>
> We conduct an ablation study by **directly incorporating chat template-formatted instruction data into the mixed pre-training stage**. We conduct the experiment on the math domain with LLaMA3-8B-Base model and evaluate its performance on the Math and Instruction Following datasets as described in Section 3.1. The results, shown in the following table, indicate that while the model's performance is comparable to our two-stage method in a few-shot setting (typically used for evaluating base models), it performs significantly worse in a zero-shot setting (commonly used for evaluating chat models). We further examine the predictions of these two methods and find that the model trained in a single stage cannot terminate its generation, often resulting in repetitive outputs. These results demonstrate that **performing all tasks in a single stage prevents the final model from effectively following human instructions**. In contrast, our proposed **two-stage mix-pretraining method successfully achieves this goal, resulting in a final chat LLM.**
>
> \\begin{array} {|l|c|c|c|}
> \\hline
> \\textrm{Model} & \\textrm{Math (Few-shot)} & \\textrm{Math (Zero-shot)} & \\textrm{Instruction Following (Zero-shot)} \\\\
> \\hline
> \\textrm{Two Stage (Ours)} &37.71 & 39.40 & 71.81 \\\\
> \\hline
> \\textrm{Single Stage} &38.83 & 8.30 & 13.64 \\\\
> \\hline
> \\end{array}
>
> **Reference**
>
> [1] Adapting large language models via reading comprehension. 2023.
>
> [2] Instruction pre-training: Language models are supervised multitask learners, 2024.

---

> ### Author Response · Authors · 2024-11-29
> **Kindly Reminder for the Discussion**
>
> Dear Reviewer 9g2u,
>
> Thanks for your careful reading of our paper. We have tried our best to elaborate the unclear points and revised our paper accordingly. We have added ablation experiments to investigate **the cost of our proposed method**, **the generation of our method to the medical domain**, **the necessity of the two-stage training method**, and **the comparison to your suggested baselines**. We would like to know whether you find our response satisfactory, or if there are more questions that we could clarify. Since the rebuttal stage is coming to an end, we are more than happy to hear your comments and address any of your further concerns during the remaining time.
>
> Best,
>
> Authors

---

> ### Author Response · Authors · 2024-11-30
> **A Friendly Reminder for Discussion**
>
> Dear Reviewer 9g2u,
>
> Thanks for your careful reading of our paper. We have added ablation experiments to investigate **the cost of our proposed method**, **the generation of our method to the medical domain**, **the necessity of the two-stage training method**, and **the comparison to your suggested baselines**. We would like to know whether you find our response satisfactory, or if there are more questions that we could clarify. Since the rebuttal stage is coming to an end, we are more than happy to hear your comments and address any of your further concerns during the remaining time.
>
> Best,
>
> Authors

---

> > ### Comment · Reviewer_9g2u · 2024-11-30
> > **Response to the authors**
> >
> > Thank you for your response. After reading other reviews and the author comments, I decided to retain my original positive feedback, marginally above the acceptance threshold.

---

> > > ### Author Response · Authors · 2024-12-01
> > > **Response to Reviewer**
> > >
> > > Dear 9g2u,
> > >
> > > Thank you for your valuable comments. We sincerely thank you for the positive reply and the continued vote to accept the work!. We are ready to address any of your further concerns during the remaining time.
> > >
> > > Thanks,
> > >
> > > Authors

---

### Official Review · Reviewer_tVX8 · 2024-11-05

**Soundness:** 2
**Presentation:** 2
**Contribution:** 2
**Rating:** 6
**Confidence:** 3

**Summary:**

In this paper, the author addresses improving the efficiency of LLMs in adapting to new knowledge. Here, efficiency refers to adapting the LLM without relying solely on instruction-based tuning for the new domain; instead, the author employs a combination of domain-specific and general data. Additionally, alignment is achieved during the pretraining stage through a unified format and an additional training objective.

**Strengths:**

The paper is well-organized, and the problem it addresses is substantial. Improving the efficiency of knowledge distillation is indeed a relevant challenge that needs attention. Furthermore, the experimental results appear promising.

**Weaknesses:**

1. The paper talks about mixing CPT, SFT, and DPO data into a combined dataset for pretraining, but mixing instruction-tuning and pretraining data isn’t really new. Plus, there aren’t many details on how exactly these datasets are blended—like what proportions are used. The approach feels a bit empirical, but it would be great to have more specifics for reproducibility.

2. The author claims that the LLM learns new knowledge efficiently by understanding the task it’s meant to perform on the data, along with alignment during pretraining. I get the alignment part, but I’m a bit concerned that task-specific instruction data might limit the generalization of domain-specific data. I wonder if it’s always beneficial for new knowledge to align closely with existing world knowledge. While task performance in the new domain improves, it’s unclear if the model’s general ability is preserved—this wasn’t really explored in the experiments.

**Questions:**

See weakness.

And I do have another question. What’s the fundamental difference between doing all of this during pretraining versus handling each step separately? Does this suggest that there’s no need to separate these steps and that it might be better to combine everything in a single pretraining process for new domain knowledge?

---

> ### Author Response · Authors · 2024-11-21
> **Response to the Concerns of Reviewer [Part 1]**
>
> Thanks for your insightful suggestions and we have listed our response to your concerns as follows. If you also have any other questions, please feel free to let us know. We will continue to try our best to answer for you.
>
> **1. The paper talks about mixing CPT, SFT, and DPO data into a combined dataset for pretraining, but mixing instruction-tuning and pretraining data isn’t really new. Plus, there aren’t many details on how exactly these datasets are blended—like what proportions are used. The approach feels a bit empirical, but it would be great to have more specifics for reproducibility.**
>
> Thank you for your valuable comments. The combination of instruction-tuning and pre-training data has been explored at an empirical level (e.g., [1]). However, as we discuss in the second paragraph of the Introduction (lines 47-53), these methods rely heavily on **specific models to synthesize large-scale, domain-specific instructions** for mixed pre-training, using **only traditional language modeling loss for training**. This reliance makes them impractical for broader domain applications and leads to significant issues with catastrophic forgetting during domain adaptation.
>
> These limitations have motivated us to propose a novel and effective framework for domain adaptation. Unlike previous approaches, we utilize readily available large-scale general instruction data (rather than domain-specific data) to **enhance the knowledge utilization capabilities** of LLMs. Additionally, our proposed logit swap self-distillation loss effectively **mitigates catastrophic forgetting**.
>
> For better reproducibility, we have detailed our pre-training and instruction-tuning data, including the number of samples and selection criteria, in Section 3.1 and Appendix A and B. We have also provided implementation details, such as evaluation settings, in Section 3.1 and Appendix D. Furthermore, after the peer review process, we will **open-source all code, including datasets, models, and training scripts**, to ensure the reproducibility of this method.
>
> **We believe our approach differs from previous work not only empirically (i.e., significant effectiveness) but also in principle and technology (i.e., a novel mixture pre-training framework and logit swap self-distillation training objective).**
>
> ————————————————————————————————————————————————————————————
>
> **2. The author claims that the LLM learns new knowledge efficiently by understanding the task it’s meant to perform on the data, along with alignment during pretraining. I get the alignment part, but I’m a bit concerned that task-specific instruction data might limit the generalization of domain-specific data. I wonder if it’s always beneficial for new knowledge to align closely with existing world knowledge. While task performance in the new domain improves, it’s unclear if the model’s general ability is preserved—this wasn’t really explored in the experiments.**
>
> Thank you for your comments. We would like to clarify that our mixture pre-training **utilizes "general" instruction data, rather than task-specific data**. This includes a wide range of scenarios such as traditional NLP tasks, daily conversations, and complex reasoning, etc. Such instruction data enhances the LLM's ability to apply memorized domain knowledge across various situations, without limiting the generalization of domain-specific data. We believe that the **knowledge utilization capacity can be essentially learned from general instruction**, which is a key distinction from prior studies that rely on domain- and task-specific instruction data that may limit their applicability and generalization.
>
> **In our experiments (refer to Tables 1 and 2)**, we demonstrate that adapting the LLM to specific domains (e.g., Wiki, Math, or Code) **improves domain performance**, while **general abilities** (e.g., English Examination, Chinese Examination, and General Domain) **remain consistent** with the original or are minimally affected. Thus, our approach optimally balances domain and general performance, achieving the best average performance.
>
> ————————————————————————————————————————————————————————————

---

> ### Author Response · Authors · 2024-11-21
> **Response to the Concerns of Reviewer [Part 2]**
>
> **3. What’s the fundamental difference between doing all of this during pretraining versus handling each step separately? Does this suggest that there’s no need to separate these steps and that it might be better to combine everything in a single pretraining process for new domain knowledge?**
>
> Thank you for your insightful comments. In the second paragraph of the Introduction (lines 41-43), we discuss that domain adaptation typically involves two stages: **knowledge learning (which includes memorization and utilization) and format alignment (responding to the user)**. The key fundamental difference in our approach is that we address these stages separately to optimize each objective individually. Furthermore, we propose that knowledge memorization and utilization can be mutually reinforced during a mixed pre-training stage. And our results in Table 1 confirm that combining these processes in **mixed pre-training outperforms traditional pre-training methods focused solely on memorization**.
>
> Despite this, the model resulting from mixed pre-training is **only a base model**. Additional instruction tuning and alignment stages are necessary to achieve the format alignment objective, essential for developing a chat model capable of human interaction. **This involves specialized chat templates that cannot be effectively learned during the pre-training phase**. Our alignment stage can be implemented more efficiently with fewer samples than previous methods.
>
> We conduct an ablation study by directly incorporating chat template-formatted instruction data into the mixed pre-training stage. We conduct the experiment on the math domain with LLaMA3-8B-Base model and evaluate its performance on the Math and Instruction Following datasets as described in Section 3.1. The results, shown in the following table, indicate that while the model's performance is comparable to our two-stage method in a few-shot setting (typically used for evaluating base models), it performs significantly worse in a zero-shot setting (commonly used for evaluating chat models). We further examine the predictions of these two methods and find that the model trained in a single stage cannot terminate its generation, often resulting in repetitive outputs. These results demonstrate that **performing all tasks in a single stage prevents the final model from effectively following human instructions**. In contrast, our proposed **two-stage mix-pretraining method successfully achieves this goal, resulting in a final chat LLM.**
>
> \\begin{array} {|l|c|c|c|}
> \\hline
> \\textrm{Model} & \\textrm{Math (Few-shot)} & \\textrm{Math (Zero-shot)} & \\textrm{Instruction Following (Zero-shot)} \\\\
> \\hline
> \\textrm{Two Stage (Ours)} &37.71 & 39.40 & 71.81 \\\\
> \\hline
> \\textrm{Single Stage} &38.83 & 8.30 & 13.64 \\\\
> \\hline
> \\end{array}
>
> **Reference**
>
> [1] Instruction-tuned Language Models are Better Knowledge Learners. Jiang, Z., Sun, Z., Shi, W., Rodriguez, P., Zhou, C., Neubig, G., Lin, X.V., Yih, W., & Iyer, S. (2024). Instruction-tuned Language Models are Better Knowledge Learners. Annual Meeting of the Association for Computational Linguistics.

---

> > ### Comment · Reviewer_tVX8 · 2024-11-26
> >
> > Thank you for your rebuttal and your clarification for my concerns. I will increase my score.

---

> > > ### Author Response · Authors · 2024-11-27
> > > **Appreciate your updated score!**
> > >
> > > Dear Reviewer tVX8,
> > >
> > > We sincerely thank you for the updated score.
> > >
> > > Thanks for your constructive review. Your review really helped us greatly in improving our paper, and we are truly grateful for your comments. We are very happy to see that you raised the score for our updated revision. And we sincerely appreciate your time and effort in reviewing our paper and reading our comments. If you also have any other questions, please feel free to let us know. We will continue to try our best to answer for you.
> > >
> > > Best,
> > >
> > > Authors

---

### Meta-Review · Area_Chair_8baT · 2024-12-20

**Metareview:**

The paper presents Mix-CPT that conducts a knowledge mixture continual pre-training that concurrently focuses on knowledge memorization and utilization. A logit swap self-distillation constraint is used to avoid catastrophic forgetting, then performs format alignment through instruction tuning and alignment.

1. The problem it addresses is substantial. [Reviewer tVX8]
2. The logit swap self-distillation constraint is a novel contribution.
3. Novel view of CPT that separates domain knowledge learning from format alignment, potentially enhances both domain adaptation efficiency and task-specific performance.Integrating domain-specific raw data with general instructional data during the pre-training phase is a strength, as it appears to balance both domain-specific and general knowledge effectively. [Reviewer wQCa].
4. The method outperforms all baselines (in most data sets in domain specific settings) [All reviewers].

Weaknesses:
1. Wikipedia is not a domain specific data, and is not a relevant data set for performance study.
2. As indicated by reviewer Yvdt, the domain specific adaptation is not better than baselines in USMLE and PubMedQA data sets.

**Additional Comments On Reviewer Discussion:**

The method is novel and all reviewers agree that the authors have addressed most of their concerns and that proposed algorithm outperforms model performance on most data sets in domain specific setting.

---

### Decision · Program_Chairs · 2025-01-22

Accept (Poster)